# Learning Nash Equilibria in Normal-Form Games via Approximating Stationary Points

## Abstract

Nash equilibrium (NE) plays an important role in game theory. However, learning an NE in normal-form games (NFGs) is a complex, non-convex optimization problem. Deep Learning (DL), the cornerstone of modern artificial intelligence, has demonstrated remarkable empirical performance across various applications involving non-convex optimization. However, applying DL to learn an NE poses significant difficulties since most existing loss functions for using DL to learn an NE introduce bias under sampled play. A recent work proposed an unbiased loss function. Unfortunately, it suffers from high variance, which degrades the convergence rate. Moreover, learning an NE through this unbiased loss function entails finding a global minimum in a non-convex optimization problem, which is inherently difficult. To improve the convergence rate by mitigating the high variance associated with the existing unbiased loss function, we propose a novel loss function, named Nash Advantage Loss (NAL). NAL is unbiased and exhibits significantly lower variance than the existing unbiased loss function. In addition, an NE is a stationary point of NAL rather than having to be a global minimum, which improves the computational efficiency. Experimental results demonstrate that the algorithm minimizing NAL achieves significantly faster empirical convergence rates compared to previous algorithms, while also reducing the variance of estimated loss value by several orders of magnitude.

## 1 Introduction

Game theory is a powerful tool for modeling multi-agent interactions. A common goal to address games is Nash equilibrium (NE) where no player gains by unilaterally deviating from this equilibrium. However, learning an NE often involves a complex, non-convex optimization problem. Theoretically, learning an NE is PPAD-complete and thus computationally intractable (Daskalakis et al., 2009).

Deep Learning (DL) (LeCun et al., 2015) has risen as a predominant technology in contemporary artificial intelligence, demonstrating remarkable performance in diverse real-world applications involving non-convex optimization problems such as image and speech recognition (Deng et al., 2014), natural language processing (Achiam et al., 2023), autonomous vehicles (Bojarski et al., 2016), and financial modeling (Heaton et al., 2017). Since NE learning is known as a non-convex optimization problem (Gemp et al., 2024), leveraging DL for NE learning presents a promising research direction. However, the application of DL to NE learning remains largely unexplored.

A significant challenge for applying DL to learn an NE is the design of an appropriate loss function. Specifically, for an $n$-player, $m$-action, general-sum normal-form game (NFG), storing the payoff matrix requires $nm^n$ entries. As $m$ and $n$ increase, the storage complexity $O(nm^n)$ grows exponentially, rendering it computationally prohibitive to load the entire payoff matrix into memory when solving large-scale NFGs. Thus, sampling a portion of the payoff matrix becomes necessary for solving large-scale NFGs. However, most existing loss functions introduce bias (Nikaidô & Isoda, 1955; Shoham & Leyton-Brown, 2008; Raghunathan et al., 2019; Gemp et al., 2022; Duan et al., 2023) under sampled play (Gemp et al., 2024), making it infeasible to learn an NE in sampled settings. To address this issue, Gemp et al. (2024) propose a novel loss function that can be unbiasedly estimated under sampled play. However, this loss function suffers from high variance as its value is estimated with the inner product of two independent and identically distributed random variables, which may degrade the convergence rate.

To improve the convergence rate by mitigating the high variance associated with the existing unbiased loss function, we propose a novel loss function called Nash Advantage Loss (NAL). Our key insight is that: finding a way to obtain an unbiased estimate of the first-order gradient to eliminate the need for calculating the inner product, which introduces high variance. Specifically, previous works overlook the fact that optimizers commonly used in DL (Robbins & Monro, 1951; Bottou, 2010; Kingma & Ba, 2014) require only unbiased estimates of the first-order gradient, rather than unbiased estimates of the loss function. Consequently, NAL guarantees that obtaining an unbiased estimate of its first-order gradient does not require the computation of the inner product between two random variables, avoiding the high variance associated with inner products. In addition, inspired by the fact that learning a stationary point (*e.g.*, a point where the first-order gradient is 0) is simpler than a global minimum since a global minimum is necessarily a stationary point while a stationary point is not always a global minimum (Jin et al., 2017), we ensure that an NE is a stationary point of NAL to improve the computational efficiency.

We conduct an empirical evaluation of the convergence rates and the variances of the estimated values of the loss functions on eight NFGs from OpenSpiel (Lanctot et al., 2019) and GAMUT (Nudelman et al., 2004). Our results reveal that the algorithm that minimizes NAL significantly surpasses the convergence rates of previous algorithms, including algorithms that minimize existing unbiased or biased loss functions. Additionally, our algorithm exhibits significantly lower variance in the estimated values of its loss function compared to the algorithm minimizing the existing unbiased loss function. Particularly, compared to the existing unbiased loss function, the variance of estimating the value of NAL is typically reduced by two orders of magnitude. In some games, this variance reduction can even reach six orders of magnitude. Moreover, we analyze the difference between the estimated and true values across different loss functions. Our findings indicate that the difference between the estimated and true values for our loss function is usually two orders of magnitude smaller compared to that of other tested loss functions.

In conclusion, our contributions are as follows:

- We propose a novel loss function for using DL to learn an NE, named Nash Advantage Loss (NAL). NAL can be estimated without bias under sampled play. More importantly, NAL will incur significantly lower variance than the existing unbiased loss function as NAL avoids the inner product of two random variables. In addition, learning an NE via minimizing NAL implies only leaning a stationary point of NAL rather than learning a global minimum.

- We conduct a comprehensive empirical evaluation of the convergence rates and the variances of estimating loss function values. The results demonstrate that our algorithm significantly outperforms the algorithms minimizing previous loss functions, in terms of the empirical convergence rate and the variance.

## 2 RELATED WORK

Our research aligns with studies that conceptualize the problem of learning an NE in NFGs as a non-convex optimization problem and address it through DL methodologies, due to DL's remarkable empirical performance in solving such problems (Chen et al., 2019; Zou et al., 2019). Specifically, we focus on studies that reduce NE computation to minimize a loss function via DL.

Sampling is critical for solving large-scale NFGs since the shape of the payoff matrix increases exponentially as the action size increases linearly. However, most existing loss functions are unsuitable for unbiased estimation under sampled play. These functions are biased under sampled play due to either (i) the presence of a random variable as the argument of a complex, nonlinear function, or (ii) unclear sampling methods (Gemp et al., 2024). For instance, duality gap based loss functions (Nikaidô & Isoda, 1955; Shoham & Leyton-Brown, 2008; Duan et al., 2023; Gemp et al., 2022) introduce bias through a max operator. Additionally, Gradient-based Nash Iteration (NI) (Raghunathan et al., 2019) is biased due to the requirement of a projection operator that projects a random variable onto the simplex, which involves a max operator (Chen & Ye, 2011). Moreover, unconstrained optimization methods (Shoham & Leyton-Brown, 2008) that penalize deviation from the simplex lose the ability to sample from strategies when each iterate is no longer within the simplex.

To mitigate bias under sampled play, Gemp et al. (2024) propose a loss function that allows unbiased estimation under sampled play. Nevertheless, this loss function suffers from high variance due to that

its value is estimated with the inner product of two independent and identically distributed random variables. Specifically, the variance of this loss function, estimated using the inner product of two random variables, is the square of the variance of estimating an individual random variable. This high variance may degrade the convergence rate. To improve the convergence rate by mitigating the high variance associated with the existing unbiased loss function, we propose a novel loss function, which allows unbiased estimation under sampled play, while incurring lower variance.

We do not consider algorithms that replicate tabular methods with DL, i.e., those that approximate table-represented variables using deep neural networks without modifying update rules, such as NFSP (Heinrich & Silver, 2016), PSRO (Lanctot et al., 2017), and Deep CFR (Brown et al., 2019). More discussions about learning NE via DL can be found in Appendix A.

## 3 PRELIMINARIES

**Normal-form games** (NFG) is a fundamental game in game theory (Osborne et al., 2004), which consists of players $\mathcal{N} = \{1, 2, \ldots, n\}$, an action set $\mathcal{A}_i$ for each player $i$, and a utility function $u_i$ for each player $i$. Each player $i \in \mathcal{N}$ simultaneously chooses an action $a_i \in \mathcal{A}_i$ and receives a utility $u_i(a_i, a_{-i}) \in [0, 1]$, where $-i$ denotes all players except player $i$. The strategy of player $i$ is represented by $\boldsymbol{x}_i \in \boldsymbol{\mathcal{X}}_i$, and the strategy profile for all players is denoted as $\boldsymbol{x} = \{\boldsymbol{x}_i \in \boldsymbol{\mathcal{X}}_i \mid i \in \mathcal{N}\}$, where $\boldsymbol{\mathcal{X}}_i$ is a $(|\mathcal{A}_i| - 1)$-dimensional simplex. The strategy space of all players are represented by $\boldsymbol{\mathcal{X}} = \times_{i \in \mathcal{N}} \boldsymbol{\mathcal{X}}_i$. Moreover, the interior of $\boldsymbol{\mathcal{X}}$ is denote $\boldsymbol{\mathcal{X}}^\circ$. Precisely, for each $\boldsymbol{x} \in \boldsymbol{\mathcal{X}}^\circ$, $\boldsymbol{x}_i(a_i) > 0, \forall i \in \mathcal{N}$ and $a_i \in \mathcal{A}_i$. The utility of player $i$, given that all players follow the strategy profile $\boldsymbol{x} \in \boldsymbol{\mathcal{X}}$, is $u_i(\boldsymbol{x}_i, \boldsymbol{x}_{-i}) = \sum_{\boldsymbol{a} \in \times_{i \in \mathcal{N}} \mathcal{A}_i} u_i(\boldsymbol{a}) \Pi_{i \in \mathcal{N}} \boldsymbol{x}_i(a_i)$, where $a_i \in \mathcal{A}_i$ denotes player $i$'s component of the joint action $\boldsymbol{a} \in \times_{i \in \mathcal{N}} \mathcal{A}_i$.

**Nash equilibrium** (NE) describes a rational behavior where no player can benefit by unilaterally deviating from the equilibrium. In other words, each player's strategy is the best response to the strategies of the other players. As analyzed in Facchinei (2003), if the utility function of each player $i$ is concave over $\boldsymbol{\mathcal{X}}_i$, an NE $\boldsymbol{x}^*$ is that $\langle \nabla_{\boldsymbol{x}_i^*} u_i(\boldsymbol{x}^*), \boldsymbol{x}_i - \boldsymbol{x}_i^* \rangle \leq 0, \forall i \in \mathcal{N}$ and $\boldsymbol{x} \in \boldsymbol{\mathcal{X}}$. This concavity condition is satisfied in NFGs since the utility function of each player $i$ is linear over $\boldsymbol{\mathcal{X}}_i$ in NFGs. We denote the set of NE by $\boldsymbol{\mathcal{X}}^*$. If the utility function of each player $i$ is concave over $\boldsymbol{\mathcal{X}}_i$, a well known metric to measure the distance from the strategy profile $\boldsymbol{x}$ to NE is the duality gap

$$\text{DualityGap}(\boldsymbol{x}) = \sum_{i \in \mathcal{N}} \max_{\boldsymbol{x}_i' \in \boldsymbol{\mathcal{X}}_i} \langle \nabla_{\boldsymbol{x}_i} u_i(\boldsymbol{x}), \boldsymbol{x}_i' - \boldsymbol{x}_i \rangle.$$

If $\text{DualityGap}(\boldsymbol{x}) = 0$, then $\boldsymbol{x} \in \boldsymbol{\mathcal{X}}^*$ and vice versa. If $\text{DualityGap}(\boldsymbol{x}) = \delta$, then $\boldsymbol{x}$ is a $\delta$-NE. We use $\boldsymbol{\mathcal{X}}^{*,\circ}$ to denote interior NE. Formally, $\forall \boldsymbol{x}^* \in \boldsymbol{\mathcal{X}}^{*,\circ}$, $\boldsymbol{x}_i^*(a_i) > 0, \forall i \in \mathcal{N}$ and $a_i \in \mathcal{A}_i$. The duality gap is the upper bound of another widely used metric, exploitability, defined as $\text{Exp}(\boldsymbol{x}) = \frac{1}{|\mathcal{N}|} \sum_{i \in \mathcal{N}} (\max_{\boldsymbol{x}_i'} u_i(\boldsymbol{x}_i', \boldsymbol{x}_{-i}) - u_i(\boldsymbol{x}_i, \boldsymbol{x}_{-i}))$. Formally, $\text{Exp}(\boldsymbol{x}) = \frac{1}{|\mathcal{N}|} \text{DualityGap}(\boldsymbol{x})$, as $u_i(\cdot)$ is linear over $\boldsymbol{\mathcal{X}}_i$ in NFGs.

**Existing unbiased loss function.** To our knowledge, the only known unbiased loss function for approximating an NE is proposed by Gemp et al. (2024). The key insight of this loss function is that gradients of all actions, w.r.t. an interior strategy profile $\boldsymbol{x} \in \boldsymbol{\mathcal{X}}^\circ$, are equal if and only if $\boldsymbol{x} \in \boldsymbol{\mathcal{X}}^*$ when the utility function of each player $i$ is concave over $\boldsymbol{\mathcal{X}}_i$. Formally, for any $\boldsymbol{x} \in \boldsymbol{\mathcal{X}}^\circ$, and $i \in \mathcal{N}$, $\nabla_{\boldsymbol{x}_i} u_i(\boldsymbol{x})(a_i) = \nabla_{\boldsymbol{x}_i} u_i(\boldsymbol{x})(a_i') \, \forall a_i, a_i' \in \mathcal{A}_i$ if and only if $\boldsymbol{x} \in \boldsymbol{\mathcal{X}}^{*,\circ}$ when the utility function $u_i(\boldsymbol{x})$ of each player $i$ is concave over $\boldsymbol{\mathcal{X}}_i$ (Gemp et al., 2024). To ensure that the interior NE always exists, they add an entropy $-\tau \boldsymbol{x}_i^{\mathrm{T}} \log \boldsymbol{x}_i$ to each player's utility function, where $\tau > 0$ is a constant. From their analysis, the addition of entropy guarantees that all equilibria of the regularization game, with utility function $u_i^\tau(\boldsymbol{x}) = u_i(\boldsymbol{x}) - \tau \boldsymbol{x}_i^{\mathrm{T}} \log \boldsymbol{x}_i$, are interior. Formally, given strategy profile $\boldsymbol{x} \in \boldsymbol{\mathcal{X}}$, their loss function is defined as follows:

$$\mathcal{L}_{Gemp}^\tau(\boldsymbol{x}) = \sum_{i \in \mathcal{N}} \|\boldsymbol{F}_i^{\tau, \boldsymbol{x}} - \overline{\boldsymbol{F}_i^{\tau, \boldsymbol{x}}}\|_2^2, \tag{1}$$

where $\boldsymbol{F}_i^{\tau, \boldsymbol{x}} = -\nabla_{\boldsymbol{x}_i} u_i^\tau(\boldsymbol{x}) = -\nabla_{\boldsymbol{x}_i} u_i(\boldsymbol{x}) + \tau \log \boldsymbol{x}_i$, and $\overline{\boldsymbol{F}_i^{\tau, \boldsymbol{x}}} = \frac{\sum_{a \in \mathcal{A}_i} \boldsymbol{F}_i^{\tau, \boldsymbol{x}}(a)}{|\mathcal{A}_i|} \mathbf{1}$. As the utility function $u_i^\tau(\cdot)$ of each player $i$ is concave over $\boldsymbol{\mathcal{X}}_i$, $\forall a_i, a_i' \in \mathcal{A}_i$, $\nabla_{\boldsymbol{x}_i} u_i^\tau(\boldsymbol{x})(a_i) = \nabla_{\boldsymbol{x}_i} u_i^\tau(\boldsymbol{x})(a_i')$ if and only if $\boldsymbol{x}$ is an NE of the regularization game. In other words, $\mathcal{L}_{Gemp}^\tau(\boldsymbol{x}) = 0$ if and only if $\boldsymbol{x}$ is an NE of the regularization game. By gradually decreasing $\tau$, the sequence of NEs of the regularization games converges to an NE of the original game. The primary advantage of this function is that it can be unbiasedly estimated given two independent unbiased estimations of $\boldsymbol{F}_i^{\tau, \boldsymbol{x}}$ (Gemp et al., 2024).

## 4 OUR METHOD

Gemp et al. (2024) propose the only known unbiased loss function for approximating an NE as defined in Eq. (1), which enables unbiased estimation under sampled play. However, this loss function often suffers from high variance, leading to considerable instability and degrading the convergence rate. Moreover, learning an NE through this loss function needs to find a global minimum in a non-convex optimization problem, a task that is inherently challenging. To improve the convergence rate by mitigating the high variance associated with the loss function defined in Eq. (1), we introduce a novel unbiased loss function called Nash Advantage Loss (NAL).

### 4.1 OVERVIEW OF NAL

Our key insight is that: finding a way to obtain an unbiased estimate of the first-order gradient to eliminate the need for calculating the inner product, which introduces high variance. In particular, the insight comes from a fact overlooked in previous work that optimizers (Robbins & Monro, 1951; Bottou, 2010; Kingma & Ba, 2014) commonly used in DL require only unbiased estimates of the first-order gradient.

**Lemma 4.1.** *For any vector $\boldsymbol{b} \in \mathbb{R}^d$ and any $\boldsymbol{y}$ in a $(d-1)$-dimensional simplex, the equation $\boldsymbol{b} - \langle \boldsymbol{b}, \boldsymbol{y} \rangle \mathbf{1} = \mathbf{0}$ holds if and only if all coordinates of $\boldsymbol{b}$ are equal.*

Specifically, NAL aims to ensure that (i) its first-order gradient can be estimated without bias using a single random variable to reduce the variance, and (ii) its first-order gradient with respect to $\boldsymbol{x} \in \boldsymbol{\mathcal{X}}^\circ$ equals $\mathbf{0}$ if and only if $\boldsymbol{x} \in \boldsymbol{\mathcal{X}}^{*,\circ}$ to improve the computational efficiency, inspired by the fact that finding a stationary point, *i.e.*, a point where the first-order gradient is 0, is simpler than finding a global minimum (Jin et al., 2017). To achieve this, we build on the key insight from the loss function defined in Eq. (1)—where for any $\boldsymbol{x} \in \boldsymbol{\mathcal{X}}^\circ$ and $i \in \mathcal{N}$, $\nabla_{\boldsymbol{x}_i} u_i(\boldsymbol{x})(a_i) = \nabla_{\boldsymbol{x}_i} u_i(\boldsymbol{x})(a_i') \,\forall a_i, a_i' \in \mathcal{A}_i$ if and only if $\boldsymbol{x} \in \boldsymbol{\mathcal{X}}^{*,\circ}$—and recognize that $\nabla_{\boldsymbol{x}_i} u_i(\boldsymbol{x})$ can be estimated without bias using a single random variable (e.g., via importance sampling). Then, inspired by Lemma 4.1, we define NAL's first-order gradient as the difference between the gradient of the utility function of the game and the inner product of the utility function's gradient with any arbitrary given strategy $\hat{\boldsymbol{x}}$. This difference is the advantage of each action's gradient for making the gradients of actions more uniform. Formally, the first-order gradient can be

$$[-\nabla_{\boldsymbol{x}_i} u_i(\boldsymbol{x}) + \langle \nabla_{\boldsymbol{x}_i} u_i(\boldsymbol{x}), \hat{\boldsymbol{x}}_i \rangle \mathbf{1} \mid i \in \mathcal{N}].$$

Since for any $\boldsymbol{x} \in \boldsymbol{\mathcal{X}}^\circ$, $i \in \mathcal{N}$ and $a_i, a_i' \in \mathcal{A}_i$, $\nabla_{\boldsymbol{x}_i} u_i(\boldsymbol{x})(a_i) = \nabla_{\boldsymbol{x}_i} u_i(\boldsymbol{x})(a_i')$ if and only if $\boldsymbol{x} \in \boldsymbol{\mathcal{X}}^{*,\circ}$, from Lemma 4.1, we have that for any $\boldsymbol{x} \in \boldsymbol{\mathcal{X}}^\circ$, $[-\nabla_{\boldsymbol{x}_i} u_i(\boldsymbol{x}) + \langle \nabla_{\boldsymbol{x}_i} u_i(\boldsymbol{x}), \hat{\boldsymbol{x}}_i \rangle \mathbf{1} \mid i \in \mathcal{N}] = \mathbf{0}$ if and only if $\boldsymbol{x} \in \boldsymbol{\mathcal{X}}^{*,\circ}$. In addition, to ensure that the interior NE always exists, as did in Gemp et al. (2024), we add an entropy $-\tau \boldsymbol{x}_i^{\mathrm{T}} \log \boldsymbol{x}_i$ to each player's utility, where $\tau > 0$ is a constant. As we mentioned above, Gemp et al. (2024) show that the additional entropy guarantees that all NE of the regularization game, with utility function $u_i^\tau(\boldsymbol{x}) = u_i(\boldsymbol{x}) - \tau \boldsymbol{x}_i^{\mathrm{T}} \log \boldsymbol{x}_i$, are interior.

Now, we provide the formal definition of NAL. Given a strategy profile $\boldsymbol{x} \in \boldsymbol{\mathcal{X}}$, NAL is defined as

$$\mathcal{L}_{NAL}^\tau(\boldsymbol{x}) = \sum_{i \in \mathcal{N}} \langle sg[\boldsymbol{F}_i^{\tau,\boldsymbol{x}} - \langle \boldsymbol{F}_i^{\tau,\boldsymbol{x}}, \hat{\boldsymbol{x}}_i \rangle \mathbf{1}], \boldsymbol{x}_i \rangle, \tag{2}$$

where $\hat{\boldsymbol{x}} = [\hat{\boldsymbol{x}}_0, \hat{\boldsymbol{x}}_1, \cdots, \hat{\boldsymbol{x}}_{|\mathcal{N}|-1}]$ can be any strategy profile within the the strategy space $\boldsymbol{\mathcal{X}}$, $\boldsymbol{F}_i^{\tau,\boldsymbol{x}} = -\nabla_{\boldsymbol{x}_i} u_i^\tau(\boldsymbol{x}) = -\nabla_{\boldsymbol{x}_i} u_i(\boldsymbol{x}) + \tau \log \boldsymbol{x}_i$ is defined in Eq. (1), and $sg[\cdot]$ is the stop-gradient operator that implies the term in this operator is not involved in gradient backpropagation. That is, in Eq. (2), $\boldsymbol{x}_i$ participates in gradient backpropagation, whereas $sg[\boldsymbol{F}_i^{\tau,\boldsymbol{x}} - \langle \boldsymbol{F}_i^{\tau,\boldsymbol{x}}, \hat{\boldsymbol{x}}_i \rangle \mathbf{1}]$ do not.

It is worth emphasizing that while other loss functions do not include the stop-gradient operator in their definitions, in practice, these loss functions must employ the stop-gradient operator when solving real-world games. This is because $F_i^{\tau,x}$ cannot feasibly participate in gradient backpropagation. Enabling $F_i^{\tau,x}$ to participate in backpropagation would require iterating over all action pairs for every two players, as done in Gemp et al. (2022) and Gemp et al. (2024), which is practically infeasible in real-world games. More details about the implementation of other loss functions are in Appendix D.

Since $sg[\boldsymbol{F}_i^{\tau,\boldsymbol{x}} - \langle \boldsymbol{F}_i^{\tau,\boldsymbol{x}}, \hat{\boldsymbol{x}}_i \rangle \mathbf{1}]$ are not involved in gradient backpropagation, we obtain

$$\nabla_{\boldsymbol{x}_i} \mathcal{L}_{NAL}^\tau(\boldsymbol{x}) = \boldsymbol{F}_i^{\tau,\boldsymbol{x}} - \langle \boldsymbol{F}_i^{\tau,\boldsymbol{x}}, \hat{\boldsymbol{x}}_i \rangle \mathbf{1}. \tag{3}$$

Since $\hat{\boldsymbol{x}}_i$ in Eq. (3) can be any strategy, not just $\boldsymbol{x}_i$, we are free to employ any sampling strategy to estimate $\nabla_{\boldsymbol{x}_i} \mathcal{L}_{NAL}^\tau(\boldsymbol{x})$ in order to reduce the variance of estimating $\nabla_{\boldsymbol{x}_i} \mathcal{L}_{NAL}^\tau(\boldsymbol{x})$.

**Unbiased estimation of NAL.**    Assume we can obtain an unbiased estimate of $\boldsymbol{F}_i^{\tau,\boldsymbol{x}}$, which can be achieved through importance sampling, as described in Section 4.3. Since $\boldsymbol{F}_i^{\tau,\boldsymbol{x}}$ is estimated without bias, $\langle \boldsymbol{F}_i^{\tau,\boldsymbol{x}}, \hat{\boldsymbol{x}}_i \rangle$ also remains unbiased, given that $\hat{\boldsymbol{x}}_i$ is known. As a result, we obtain an unbiased estimate of $\nabla_{\boldsymbol{x}_i}\mathcal{L}_{NAL}^\tau(\boldsymbol{x})$ using the unbiased estimates of $\boldsymbol{F}_i^{\tau,\boldsymbol{x}}$ and $\langle \boldsymbol{F}_i^{\tau,\boldsymbol{x}}, \hat{\boldsymbol{x}}_i \rangle$. Then, with the unbiased estimate of $\nabla_{\boldsymbol{x}_i}\mathcal{L}_{NAL}^\tau(\boldsymbol{x})$ and the known $\dot{\boldsymbol{x}}_i$, an unbiased estimate of $\mathcal{L}_{NAL}^\tau(\boldsymbol{x})$ is obtained. This unbiased estimate of $\mathcal{L}_{NAL}^\tau(\boldsymbol{x})$ is passed to the optimizer to update the parameters of the deep neural network. Further details on the unbiased estimation process can be found in Section 4.3.

**Relationship between duality gap in the regularization game and NAL.**    As analyzed in Gemp et al. (2024), $\forall a_i, a_i' \in \mathcal{A}_i$, $\nabla_{\boldsymbol{x}_i}u_i^\tau(\boldsymbol{x})(a_i) = \nabla_{\boldsymbol{x}_i}u_i^\tau(\boldsymbol{x})(a_i')$ holds if and only if $\boldsymbol{x}$ is an NE of the regularization game with the utility function $u_i^\tau(\boldsymbol{x})$. Then, from Lemma 4.1, $\nabla_{\boldsymbol{x}}\mathcal{L}_{NAL}^\tau(\boldsymbol{x}) = \boldsymbol{0}$ if and only if $\boldsymbol{x}$ is an NE of the regularization game with the utility function $u_i^\tau(\boldsymbol{x})$. Combining these, we conclude that a stationary point of NAL, where $\nabla_{\boldsymbol{x}}\mathcal{L}_{NAL}^\tau(\boldsymbol{x}) = \boldsymbol{0}$, is necessarily an NE of the regularization game with utility function $u_i^\tau(\boldsymbol{x})$, and vice versa. This is because a global minimum is always a stationary point, whereas a stationary point is not necessarily a global minimum (Jin et al., 2017). A formal relationship between the duality gap of a strategy profile $\boldsymbol{x}$ in the regularization game and the gradient of NAL is in Theorem 4.2. The proof of Theorem 4.2 depends on the properties of the tangent residual (Cai et al., 2022).

**Theorem 4.2.** *(Proof is in Appendix B.2) The duality gap of a given strategy profile $\boldsymbol{x}$ in the regularization game, with the utility function $u_i^\tau(\boldsymbol{x}) = u_i(\boldsymbol{x}) - \tau \boldsymbol{x}_i^{\mathrm{T}}\log\boldsymbol{x}_i$, is upper bounded as*

$$DualityGap^\tau(\boldsymbol{x}) = \sum_{i\in\mathcal{N}} \max_{\boldsymbol{x}_i'\in\mathcal{X}_i} \langle \nabla_{\boldsymbol{x}_i}u_i^\tau(\boldsymbol{x}), \boldsymbol{x}_i' - \boldsymbol{x}_i \rangle \le C_0 \|\nabla_{\boldsymbol{x}}\mathcal{L}_{NAL}^\tau(\boldsymbol{x})\|_2,$$

*where $C_0$ is a game-dependent constant.*

Note that the exploitability in the regularization game $\mathrm{Exp}^\tau(\boldsymbol{x}) = \frac{1}{|\mathcal{N}|}\sum_{i\in\mathcal{N}}(\max_{\boldsymbol{x}_i'} u_i^\tau(\boldsymbol{x}_i', \boldsymbol{x}_{-i}) - u_i^\tau(\boldsymbol{x}_i, \boldsymbol{x}_{-i})) \le \frac{1}{|\mathcal{N}|}DualityGap^\tau(\boldsymbol{x})$ since the function $u_i(\boldsymbol{x}_i)$ and $-\tau\boldsymbol{x}_i^{\mathrm{T}}\log\boldsymbol{x}_i$ for each player $i$ are linear and concave over $\mathcal{X}_i$, respectively, and for any concave function $f(\cdot)$ with any $\boldsymbol{u}, \boldsymbol{v}$ in its domain, the inequality $f(\boldsymbol{u}) - f(\boldsymbol{v}) \le \langle \nabla f(\boldsymbol{v}), \boldsymbol{u} - \boldsymbol{v} \rangle$ holds.

**Relationship between duality gap in the original game and NAL.**    NAL ensures that a stationary point of NAL corresponds to an NE of the regularization game rather than the original game. To find an NE of the original game, we establishes a precise relationship between the duality gap in the original game and NAL, as shown in Theorem 4.3. This relationship allows us to approximate an NE of the original game by minimizing NAL. Specifically, by progressively decreasing the value of $\tau$, we guarantee that the sequence of NE of the regularization games, characterized by the utility function $u_i^\tau(\boldsymbol{x}) = u_i(\boldsymbol{x}) - \tau$, converges to an NE of the original game.

**Theorem 4.3.** *(Proof is in Appendix B.4) The duality gap of a given strategy profile $\boldsymbol{x}$ in the original game is upper bounded as:*

$$DualityGap(\boldsymbol{x}) \le \tau C_1 + C_2\|\nabla_{\boldsymbol{x}}\mathcal{L}_{NAL}^\tau(\boldsymbol{x})\|_2,$$

*where $C_1$ and $C_2$ are game-dependent constants.*

## 4.2 Analysis of Variance of NAL and Existing Unbiased Loss Function

We now analyze the variance in the estimated values of NAL and the unbiased loss function defined in Eq. (1). We demonstrate that when the variance in estimating the value of NAL is $O(\sigma)$, that of the unbiased loss function defined in Eq. (1), may be $O(\sigma^2)$, where $\sigma > 0$ is a constant.

Firstly, assume that the components of the vector $\boldsymbol{F}_i^{\tau,\boldsymbol{x}} - \langle \boldsymbol{F}_i^{\tau,\boldsymbol{x}}, \hat{\boldsymbol{x}}_i \rangle \boldsymbol{1}$ at each $\boldsymbol{a}_i \in \mathcal{A}_i$ are estimated independently, with the variance for each estimation is less than $\sigma$. Specifically, let the estimation of $\boldsymbol{F}_i^{\tau,\boldsymbol{x}} - \langle \boldsymbol{F}_i^{\tau,\boldsymbol{x}}, \hat{\boldsymbol{x}}_i \rangle \boldsymbol{1}$ at action $\boldsymbol{a}_i \in \mathcal{A}_i$ be denoted as $\hat{\boldsymbol{g}}_i^{\tau,\boldsymbol{x}}(a_i)$. Under this assumption, we have $\hat{\boldsymbol{g}}_i^{\tau,\boldsymbol{x}}(a_i) \perp \hat{\boldsymbol{g}}_i^{\tau,\boldsymbol{x}}(a_i')$, where $\perp$ denotes that the two random variables are independent, and $\mathrm{Var}[\hat{\boldsymbol{g}}_i^{\tau,\boldsymbol{x}}(a_i)] \le \sigma$ for all $a_i, a_i' \in \mathcal{A}_i$. By the definition of variance, the variance of $\mathcal{L}_{NAL}^\tau(\boldsymbol{x})$ is

$$\mathrm{Var}[\mathcal{L}_{NAL}^\tau(\boldsymbol{x})] = \sum_{i\in\mathcal{N}}\sum_{a_i\in\mathcal{A}_i}\mathrm{Var}[\hat{\boldsymbol{g}}_i^{\tau,\boldsymbol{x}}(a_i)\boldsymbol{x}_i(a_i)] = \sum_{i\in\mathcal{N}}\sum_{a_i\in\mathcal{A}_i}(\boldsymbol{x}_i(a_i))^2\mathrm{Var}[\hat{\boldsymbol{g}}_i^{\tau,\boldsymbol{x}}(a_i)] \le |\mathcal{N}|\sigma,$$

where the second equality comes from the fact that for a random variable $Y$ with a constant $c$, $\mathrm{Var}[cY] = c^2\mathrm{Var}[Y]$, and the inequality follows from the fact that $\sum_{a_i\in\mathcal{A}_i}(\boldsymbol{x}_i(a_i))^2 \le 1$.

---

**Algorithm 1** Learning an NE via Minimizing NAL

---

1: **Input:** An optimizer $\mathcal{OPT}$, the exploration ratio $\epsilon$, the uniform strategy profile $\boldsymbol{x}^u = [\boldsymbol{x}_i^u | i \in \mathcal{N}]$, the initial parameter $\boldsymbol{\theta}$, the learning rate $\eta$, the regularization scalar $\tau$, the number of total iterations $T$, the number of instances $S$ sampled at per iteration, the frequency $T_u$ of updating $\eta$ and $\tau$, the weight $\alpha$ on updating $\eta$, the weight $\beta$ on updating $\tau$, simulator $\mathcal{G}$ that returns player $i$'s payoff given a joint action.

2: **for** each $t \in [1, 2, \cdots, T]$ **do**

3:     Initialize buffer $\mathcal{M}_i \leftarrow \{\}, \forall i \in \mathcal{N}$

4:     $v_i \leftarrow 0, \forall i \in \mathcal{N}$

5:     **for** each $s \in [1, 2, \cdots, S]$ **do**

6:         $a_i \sim \boldsymbol{x}_i^{\boldsymbol{\theta}}, \forall i \in \mathcal{N}$

7:         $\boldsymbol{a} \leftarrow [a_i : i \in \mathcal{N}]$

8:         $a_i' \sim (1 - \epsilon)\boldsymbol{x}_i^{\boldsymbol{\theta}} + \epsilon \boldsymbol{x}_i^u, \forall i \in \mathcal{N}$

9:         $p_i \leftarrow (1 - \epsilon)\boldsymbol{x}_i^{\boldsymbol{\theta}}(a_i') + \epsilon \boldsymbol{x}_i^u(a_i'), \forall i \in \mathcal{N}$

10:        $r_i \leftarrow -\mathcal{G}(i, a_i', \boldsymbol{a}_{-i}) + \tau \log \boldsymbol{x}_i^{\boldsymbol{\theta}}(a_i'), \forall i \in \mathcal{N}$           $\triangleright$ To estimate $\boldsymbol{F}_i^{\tau, \boldsymbol{x}^{\boldsymbol{\theta}}}(a_i')$

11:        $\mathcal{M}_i$.append($[i, a_i', r_i, p_i]), \forall i \in \mathcal{N}$

12:        $v_i \leftarrow v_i + r_i$

13:     **end for**

14:     $\tilde{\mathcal{L}}_{NAL}^{\tau}(\boldsymbol{\theta}) \leftarrow 0$

15:     $v_i \leftarrow \frac{v_i}{S}, \forall i \in \mathcal{N}$                              $\triangleright$ To estimate $\langle \boldsymbol{F}_i^{\tau, \boldsymbol{x}^{\boldsymbol{\theta}}}, \hat{\boldsymbol{x}}_i \rangle$

16:     **for** each $i \in \mathcal{N}$ **do**

17:        **for** each $[i, a_i^s, r_i^s, p_i^s] \in \mathcal{M}_i$ **do**

18:           $\boldsymbol{g}_i^s \leftarrow \frac{r_i^s - v_i}{p_i^s} \boldsymbol{e}_{a_i^s}$                $\triangleright$ To estimate $\boldsymbol{F}_i^{\tau, \boldsymbol{x}^{\boldsymbol{\theta}}} - \langle \boldsymbol{F}_i^{\tau, \boldsymbol{x}^{\boldsymbol{\theta}}}, \hat{\boldsymbol{x}}_i \rangle \mathbf{1}$

19:           $\tilde{\mathcal{L}}_{NAL}^{\tau}(\boldsymbol{\theta}) \leftarrow \tilde{\mathcal{L}}_{NAL}^{\tau}(\boldsymbol{\theta}) + \langle \boldsymbol{g}_i^s, \dot{\boldsymbol{x}}_i^{\boldsymbol{\theta}} \rangle$

20:        **end for**

21:     **end for**

22:     $\boldsymbol{\theta} \leftarrow \mathcal{OPT}$.update($\tilde{\mathcal{L}}_{NAL}^{\tau}(\boldsymbol{\theta})$)

23:     **if** $t\%T_u = 0$ **then**

24:        $\eta \leftarrow \alpha \eta, \tau \leftarrow \beta \tau$

25:     **end if**

26: **end for**

27: **Return** $\boldsymbol{\theta}$

---

For the unbiased loss function defined in Eq. (1), we make similar assumptions. Specifically, let the two estimates of $\boldsymbol{F}_i^{\tau, \boldsymbol{x}}(a_i) - \overline{\boldsymbol{F}_i^{\tau, \boldsymbol{x}}}(a_i)$ are $\bar{\boldsymbol{g}}_i^{\tau, \boldsymbol{x}, 1}(a_i)$ and $\bar{\boldsymbol{g}}_i^{\tau, \boldsymbol{x}, 2}(a_i)$, we assume that each $\bar{\boldsymbol{g}}_i^{\tau, \boldsymbol{x}, j}(a_i)$ is sampled independently for all $i \in \mathcal{N}, a_i \in \mathcal{A}_i, j \in \{1, 2\}$, and the variance for each estimation is less than $\sigma$. Formally, for all $i \in \mathcal{N}, a_i, a_i' \in \mathcal{A}_i, j, j' \in \{1, 2\}, \bar{\boldsymbol{g}}_i^{\tau, \boldsymbol{x}, j}(a_i) \perp \bar{\boldsymbol{g}}_i^{\tau, \boldsymbol{x}, j'}(a_i')$ and $\text{Var}[\bar{\boldsymbol{g}}_i^{\tau, \boldsymbol{x}, j}(a_i)] \leq \sigma$. Then, the variance of the estimation for this loss function is

$$\text{Var}[\mathcal{L}_{Gemp}^{\tau}(\boldsymbol{x})] = \sum_{i \in \mathcal{N}} \sum_{a_i \in \mathcal{A}_i} \text{Var}[\bar{\boldsymbol{g}}_i^{\tau, \boldsymbol{x}, 1}(a_i) \bar{\boldsymbol{g}}_i^{\tau, \boldsymbol{x}, 2}(a_i)]$$

$$\leq |\mathcal{N}|\sigma^2 \max_{i \in \mathcal{N}} |\mathcal{A}_i| + 2|\mathcal{N}|\sigma \max_{i \in \mathcal{N}, a_i \in \mathcal{A}_i} \|\boldsymbol{F}_i^{\tau, \boldsymbol{x}}(a_i) - \overline{\boldsymbol{F}_i^{\tau, \boldsymbol{x}}}(a_i)\|_2^2 \max_{i \in \mathcal{N}} |\mathcal{A}_i|,$$

where the last inequality follows from Appendix C. Therefore, the variance in estimating $\mathcal{L}_{Gemp}^{\tau}(\boldsymbol{x})$ is $\sigma \max_{i \in \mathcal{N}} |\mathcal{A}_i|$ times larger than for NAL. Consequently, the variance in estimating $\mathcal{L}_{Gemp}^{\tau}(\boldsymbol{x})$ is expected to be substantially higher than that of NAL.

### 4.3 MINIMIZING NAL UNDER SAMPLED PLAY

We now detail our algorithm that learns an NE by minimizing NAL. The pseudocode is provided in Algorithm 1. Specifically, consider a deep neural network $\Pi(\cdot)$ parameterized by $\boldsymbol{\theta}$, where the resulting strategy profile is denoted as $\boldsymbol{x}^{\boldsymbol{\theta}} = \Pi(\boldsymbol{\theta})$. Our objective is to minimize the following loss function $\mathcal{L}_{NAL}^{\tau}(\boldsymbol{\theta})$ through a two-step process: **sampling** and **updating**.

$$\mathcal{L}_{NAL}^{\tau}(\boldsymbol{\theta}) = \sum_{i \in \mathcal{N}} \langle sg[\boldsymbol{F}_i^{\tau, \boldsymbol{x}^{\boldsymbol{\theta}}} - \langle \boldsymbol{F}_i^{\tau, \boldsymbol{x}^{\boldsymbol{\theta}}}, \hat{\boldsymbol{x}}_i \rangle \mathbf{1}], \boldsymbol{x}_i^{\boldsymbol{\theta}} \rangle.$$

**Sampling.** The sampling process is outlined from lines 3 to 13 in Algorithm 1. At each iteration $t$, we begin by initializing the buffer $\mathcal{M}_i = \{\}$ and the random variable $v_i$ for each player $i$ (lines 3 and 4 of Algorithm 1). The random variable $v_i$ is used to estimate the value of $-\langle \boldsymbol{F}_i^{\tau, \boldsymbol{x}^{\boldsymbol{\theta}}}, \hat{\boldsymbol{x}}_i \rangle$. Next, for each player $i$, $S$ instances are sampled. In each instance, an action $a_i$ is selected for each player $i$ according to the strategy profile $\boldsymbol{x}^{\boldsymbol{\theta}}$ (line 6 of Algorithm 1), resulting in the action

profile $\boldsymbol{a} = [a_i : i \in \mathcal{N}]$ (line 7 of Algorithm 1). Each $\boldsymbol{a}_{-i} = [a_j : j \in \mathcal{N}, j \neq i]$ serves as the environmental dynamic for player $i$, enabling the estimation of $\mathcal{L}_{NAL}^\tau(\boldsymbol{\theta})$. Subsequently, based on the exploration parameter $\epsilon$, the uniform strategy profile $\boldsymbol{x}^u$ (i.e., $\boldsymbol{x}_i^u(a_i) = \frac{1}{|\mathcal{A}_i|}$, for all $i \in \mathcal{N}$ and $a_i \in \mathcal{A}_i$), and the current strategy profile $\boldsymbol{x}^{\boldsymbol{\theta}}$, an alternative action $a_i'$ is sampled for each player $i$ according to the strategy $\hat{\boldsymbol{x}}_i = (1 - \epsilon)\boldsymbol{x}_i^{\boldsymbol{\theta}} + \epsilon \boldsymbol{x}_i^u$ (line 8 of Algorithm 1). The exploration parameter $\epsilon$ and the uniform strategy $\boldsymbol{x}^u$ ensure that the probability of selecting any action $a$ within the strategy $\hat{\boldsymbol{x}}_i$ is not too small, which guarantees the variance of estimating via importance sampling is not too large. The probability of selecting action $a_i'$ through $\hat{\boldsymbol{x}}_i$ is denoted by $p_i$ (line 9 of Algorithm 1). The unbiased estimation of $\boldsymbol{F}_i^{\tau, \boldsymbol{x}^{\boldsymbol{\theta}}}(a_i')$ is then computed as $r_i \leftarrow -\mathcal{G}(i, a_i', \boldsymbol{a}_{-i}) + \tau \log \boldsymbol{x}_i^{\boldsymbol{\theta}}(a_i')$, $\forall i \in \mathcal{N}$ (line 10 of Algorithm 1), where $\mathcal{G}$ represents the simulator returning player $i$'s payoff for the joint action $[a_i', \boldsymbol{a}_{-i}]$. Specifically,

$$
\begin{aligned}
\mathbb{E}[r_i] = \mathbb{E}[-\mathcal{G}(i, a_i', \boldsymbol{a}_{-i}) + \tau \log \boldsymbol{x}_i^{\boldsymbol{\theta}}(a_i')] &= \mathbb{E}[-\mathcal{G}(i, a_i', \boldsymbol{a}_{-i})] + \tau \log \boldsymbol{x}_i^{\boldsymbol{\theta}}(a_i') \\
&= \boldsymbol{F}_i^{\boldsymbol{x}^{\boldsymbol{\theta}}}(a_i') + \tau \log \boldsymbol{x}_i^{\boldsymbol{\theta}}(a_i') = \boldsymbol{F}_i^{\tau, \boldsymbol{x}^{\boldsymbol{\theta}}}(a_i'),
\end{aligned}
\tag{4}
$$

where the second line follows from the fact that $\boldsymbol{a}_{-i}$ is sampled according to $\boldsymbol{x}_{-i}^{\boldsymbol{\theta}}$. Finally, the tuple $[i, a_i', r_i, p_i]$ is stored in the buffer $\mathcal{M}_i$ (line 11 of Algorithm 1), and $v_i$ is updated as $v_i \leftarrow v_i + r_i$ (line 12 of Algorithm 1).

**Updating.** The updating procedure is outlined from lines 14 to 25 in Algorithm 1. We first initialize the estimator for $\mathcal{L}_{NAL}^\tau(\boldsymbol{\theta})$ as $\tilde{\mathcal{L}}_{NAL}^\tau(\boldsymbol{\theta}) \leftarrow 0$ and normalize $v_i$ by setting $v_i \leftarrow \frac{v_i}{S}$ (lines 14 and 15 of Algorithm 1). The expectation $\mathbb{E}[v_i]$ corresponds to $\langle \boldsymbol{F}_i^{\tau, \boldsymbol{x}^{\boldsymbol{\theta}}}, \hat{\boldsymbol{x}}_i \rangle$. Formally,

$$
\mathbb{E}[v_i] = \mathbb{E}\left[ \frac{1}{S} \sum_{s=1}^S r_i^s \right] = \mathbb{E}\left[ \frac{1}{S} \sum_{s=1}^S \boldsymbol{F}_i^{\tau, \boldsymbol{x}^{\boldsymbol{\theta}}}(a_i^s) \right] = \mathbb{E}_{a_i^s \sim \hat{\boldsymbol{x}}_i}\left[ \boldsymbol{F}_i^{\tau, \boldsymbol{x}^{\boldsymbol{\theta}}}(a_i^s) \right] = \langle \boldsymbol{F}_i^{\tau, \boldsymbol{x}^{\boldsymbol{\theta}}}, \hat{\boldsymbol{x}}_i \rangle,
\tag{5}
$$

where $a_i^s$ and $r_i^s$ come from the $s$-th tuple $[i, a_i^s, r_i^s, p_i^s]$ stored in buffer $\mathcal{M}_i$, the second equality follows from $\mathbb{E}[r_i^s] = \boldsymbol{F}_i^{\tau, \boldsymbol{x}^{\boldsymbol{\theta}}}(a_i^s)$ (Eq. (4)), and the third equality is from that $a_i^s$ is sampled via $\hat{\boldsymbol{x}}_i$. Additionally, we use the tuples in $\mathcal{M}_i$ (line 17 of Algorithm 1) to estimate $\boldsymbol{F}_i^{\tau, \boldsymbol{x}^{\boldsymbol{\theta}}} - \langle \boldsymbol{F}_i^{\tau, \boldsymbol{x}^{\boldsymbol{\theta}}}, \hat{\boldsymbol{x}}_i \rangle \mathbf{1}$ through the computation $\boldsymbol{g}_i^s \leftarrow \frac{r_i^s - v_i}{p_i^s} \boldsymbol{e}_{a_i^s}$ (line 18 of Algorithm 1), where $\boldsymbol{e}_{a_i^s}$ is a vector whose the coordinate $a_i^s$ is 1 and all other coordinates are 0. It is straightforward to verify that $\mathbb{E}[\boldsymbol{g}_i^s] = \boldsymbol{F}_i^{\tau, \boldsymbol{x}^{\boldsymbol{\theta}}} - \langle \boldsymbol{F}_i^{\tau, \boldsymbol{x}^{\boldsymbol{\theta}}}, \hat{\boldsymbol{x}}_i \rangle \mathbf{1}$. Formally,

$$
\mathbb{E}[\boldsymbol{g}_i^s] = \mathbb{E}_{s \sim p_i(s)}\left[ \frac{r_i^s - v_i}{p_i^s} \boldsymbol{e}_{a_i^s} \right] = \mathbb{E}_{s \sim p_i(s)}\left[ \frac{\boldsymbol{F}_i^{\tau, \boldsymbol{x}^{\boldsymbol{\theta}}}(a_i^s) - \langle \boldsymbol{F}_i^{\tau, \boldsymbol{x}^{\boldsymbol{\theta}}}, \hat{\boldsymbol{x}}_i \rangle}{p_i^s} \boldsymbol{e}_{a_i^s} \right],
\tag{6}
$$

where the second equality follows from $\mathbb{E}[r_i^s] = \boldsymbol{F}_i^{\tau, \boldsymbol{x}^{\boldsymbol{\theta}}}(a_i^s)$ (Eq. (4)), $\mathbb{E}[v_i] = \langle \boldsymbol{F}_i^{\tau, \boldsymbol{x}^{\boldsymbol{\theta}}}, \hat{\boldsymbol{x}}_i \rangle$ (Eq. (5)), and $(r_i^s - v_i) \perp p_i^s$ (since $p_i^s$ is given and not sampled, which can be seen as a constant). As the rightest side of Eq. (6) a standard importance sampling process, it follows from the properties of importance sampling that

$$
\mathbb{E}[\boldsymbol{g}_i^s] = \mathbb{E}_{s \sim p_i(s)}\left[ \frac{\boldsymbol{F}_i^{\tau, \boldsymbol{x}^{\boldsymbol{\theta}}}(a_i^s) - \langle \boldsymbol{F}_i^{\tau, \boldsymbol{x}^{\boldsymbol{\theta}}}, \hat{\boldsymbol{x}}_i \rangle}{p_i^s} \boldsymbol{e}_{a_i^s} \right] = \boldsymbol{F}_i^{\tau, \boldsymbol{x}^{\boldsymbol{\theta}}} - \langle \boldsymbol{F}_i^{\tau, \boldsymbol{x}^{\boldsymbol{\theta}}}, \hat{\boldsymbol{x}}_i \rangle \mathbf{1}.
$$

The estimator $\tilde{\mathcal{L}}_{NAL}^\tau(\boldsymbol{\theta})$ is updated via $\tilde{\mathcal{L}}_{NAL}^\tau(\boldsymbol{\theta}) \leftarrow \tilde{\mathcal{L}}_{NAL}^\tau(\boldsymbol{\theta}) + \langle \boldsymbol{g}_i^s, \dot{\boldsymbol{x}}_i^{\boldsymbol{\theta}} \rangle$ (line 19 of Algorithm 1). Since $\mathbb{E}[\boldsymbol{g}_i^s] = \boldsymbol{F}_i^{\tau, \boldsymbol{x}^{\boldsymbol{\theta}}} - \langle \boldsymbol{F}_i^{\tau, \boldsymbol{x}^{\boldsymbol{\theta}}}, \hat{\boldsymbol{x}}_i \rangle \mathbf{1}$ and $\dot{\boldsymbol{x}}_i^{\boldsymbol{\theta}}$ is known, it follows that $\frac{1}{S}\mathbb{E}[\tilde{\mathcal{L}}_{NAL}^\tau(\boldsymbol{\theta})] = \mathcal{L}_{NAL}^\tau(\boldsymbol{\theta})$. Therefore, $\tilde{\mathcal{L}}_{NAL}^\tau(\boldsymbol{\theta})$ provides an unbiased estimate of $\mathcal{L}_{NAL}^\tau(\boldsymbol{\theta})$. The estimator $\tilde{\mathcal{L}}_{NAL}^\tau(\boldsymbol{\theta})$ is then passed to the optimizer $\mathcal{OPT}$ for updating $\boldsymbol{\theta}$ (line 22 of Algorithm 1). If $t\%T_u = 0$ (line 23 of Algorithm 1), the parameters $\eta$ and $\tau$ are updated as $\eta \leftarrow \alpha\eta$ and $\tau \leftarrow \beta\tau$, where $0 < \alpha, \beta < 1$ (line 24 of Algorithm 1). These adjustments ensure that the NE of the regularization game approaches the NE of the original game. Specifically, as shown in Theorem 4.3, decreasing $\tau$ brings the NE of the regularization game, defined by the utility function $u_i^\tau(\boldsymbol{x}) = u_i(\boldsymbol{x}) - \tau \boldsymbol{x}_i^{\mathsf{T}} \log \boldsymbol{x}_i$, closer to that of the original game. Furthermore, reducing $\eta$ stabilizes the algorithm as we find that without a corresponding reduction in $\eta$, decreasing $\tau$ could destabilize the learning process.

We do not provide the convergence for our algorithm since this convergence depends on the optimizer used, which is not the focus of this work. Additionally, the introduction of DL further increases the difficulty of analysis as deep neural networks are non-smooth. Learning stationary points theoretically with such a non-convex and non-smooth function is an urgent problem to be solved. Solving this problem belongs to the research direction of optimization rather than game theory.

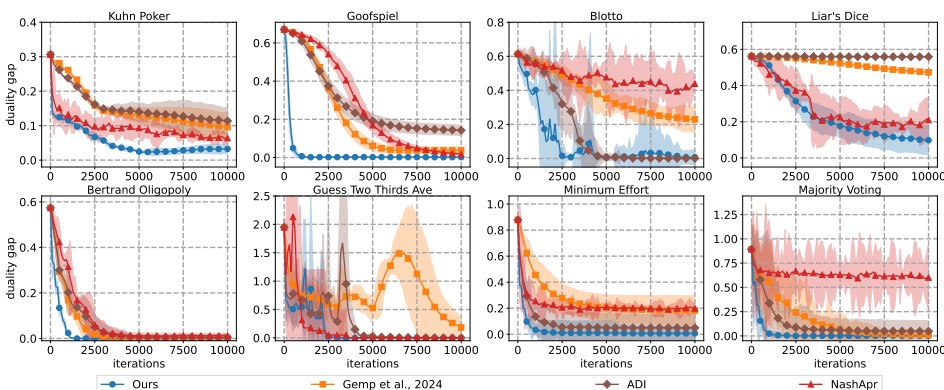

Figure 1: Empirical convergence rates of the tested algorithms when the optimizer is Adam. The top row shows the following scenarios from left to right: 2 players with 64 actions, 2 players with 384 actions, 4 players with 66 actions, and 2 players with 2304 actions. The bottom row displays, from left to right: 4 players with 50 actions, 4 players with 50 actions, 5 players with 30 actions, and 11 players with 5 actions. The shaded regions represent one standard deviation of the results, calculated across four different random seeds.

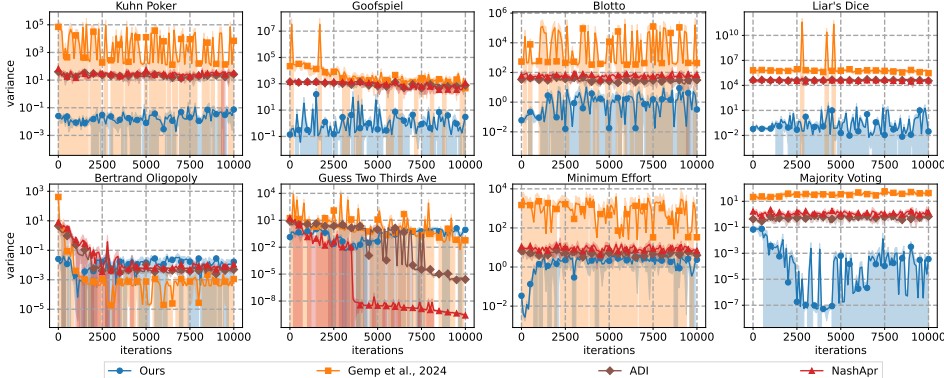

Figure 2: Variances observed in estimating the value of loss functions used by different algorithms when the optimizer is Adam.

## 5 EXPERIMENTS

**Configurations.** We compare our algorithm with algorithms that minimize the loss function proposed by Gemp et al. (2024), ADI (Gemp et al., 2022), or NashApr (Duan et al., 2023), respectively. Our loss function and the loss function provided by Gemp et al. (2024) are unbiased loss functions, while ADI and NashApr are biased loss functions. The implementation details of the compared loss functions are in Appendix D. Notably, the implementation of all tested loss functions includes the stop-gradient operator. We conduct experiments on eight NFGs from OpenSpiel (Lanctot et al., 2019) and GAMUT (Nudelman et al., 2004), specifically Kuhn Poker, Goofspiel, Blotto, Liars Dice, Bertrand Oligopoly, Guess Two Thirds Ave, Minimum Effort, and Majority Voting. The former four games are sourced from OpenSpiel, while other games are implemented by GAMUT. The payoff matrix components of each game are normalized to a range between 0 and 1. All experiments are performed on a machine equipped with four RTX 3060 GPUs and 376 GB of memory.

The network, parameterized by $\boldsymbol{\theta}$ and responsible for representing strategy profiles, is structured as a three-layer MLP. Both the input and hidden layers consist of 1024 neurons, while the output layer has $|\mathcal{N}|$ heads, where each head's dimension corresponds to the action space of its respective player. The hidden layers utilize the ReLU activation function (Krizhevsky et al., 2012), and the output layer applies the Softmax activation function (Dempster et al., 1977), ensuring that the output resides within the simplex. To reduce computational cost, a single parameter update is applied per iteration.

For all algorithms tested, the value of $\epsilon$ is fixed at 1. The parameter $T$ is set to 10,000, while $S$ is fixed at 10 across all games. Neural networks and optimizers are implemented using PyTorch (Paszke et al., 2019). We utilize Adam (Kingma & Ba, 2014) as the optimizer, given its widespread adoption

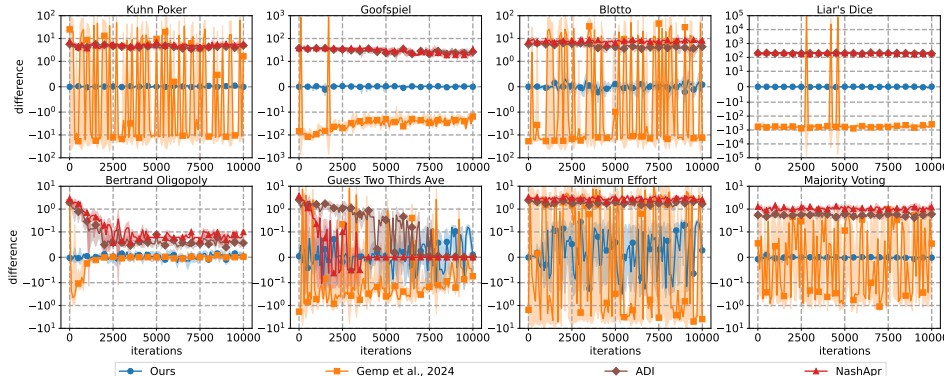

Figure 3: Difference between the true value and the estimated value of loss functions when the optimizer is Adam. Since the difference between the true value and the estimated value of our loss function NAL is considerably smaller than that of other loss functions, we present a more detailed graph highlighting this difference for NAL in Appendix E.

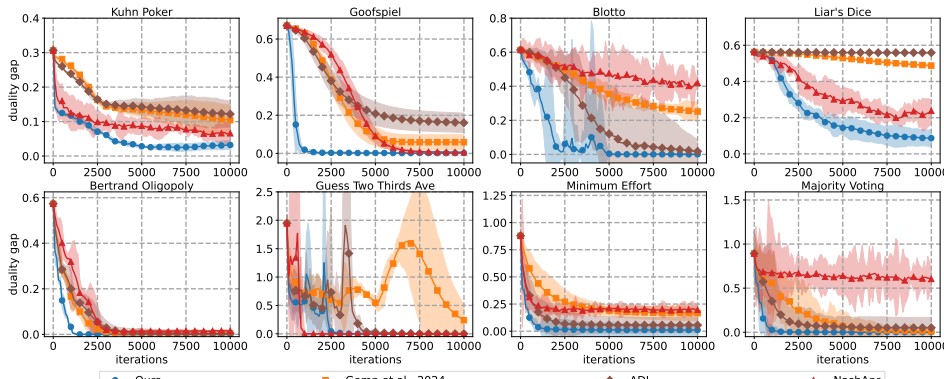

Figure 4: Empirical convergence rates of the tested algorithms when the optimizer is RMSprop.

in training modern deep neural networks, including GANs (Goodfellow et al., 2014), BERT (Devlin, 2018), GPT (Brown, 2020), and ViT (Dosovitskiy, 2020). To optimize the convergence performance of the evaluated algorithms with Adam, we conducted an extensive hyperparameter search. Specifically, we varied the learning rate $\eta \in \{0.0001, 0.00001\}$, the regularization scalar $\tau \in \{0.1, 1\}$, the update frequency $T_u \in \{200, 500, 1000\}$, the weight $\alpha \in \{0.9, 0.5\}$, and the weight $\beta \in \{0.9, 0.5\}$. The selected hyperparameters are shown in Appendix F.

**Results on convergence rates and variances.** We run each algorithm four times with different random seeds for each run. The results, including convergence rates and variances, are presented in Figures 1 and 2, respectively. Notably, our algorithm achieves the fastest empirical convergence and the lowest variance across all evaluated algorithms. Specifically, the variance in estimating NAL decreases by at least two orders of magnitude for all tested games compared to using the existing unbiased loss function, and in Liars Dice, this variance reduction reaches up to six orders of magnitude. In addition, we find that, in Goofspiel and Minimum Effort, the algorithm minimizing the existing unbiased loss function defined in Eq. (1) fails to converge to an NE. In contrast, the algorithm minimizing NAL to learn an NE is able to converge to an NE. Additionally, algorithms based on biased loss functions occasionally fail to converge. For example, the algorithm minimizing ADI does not converge in Blotto, and the algorithm minimizing NashApr fails in Liars Dice. We also find a strong correlation between variance and convergence performance. In Bertrand Oligopoly, where the algorithm minimizing the existing unbiased loss function defined in Eq. (1) performs closest to ours, it is the only case where this algorithm's variance in estimating the value of the loss function is lower than that of our algorithm. However, due to the extremely high variance early on, this algorithm's convergence rate remains slower than ours. Although the algorithm minimizing NAL does not appear to converge to an exact NE in Kuhn Poker and Liars Dice, this is primarily due to the value of $S$ being insufficiently large. When $S$ is increased, the algorithm minimizing NAL can also converge to a more and more accurate NE in both Kuhn Poker and Liars Dice. More details can be found in Appendix E.

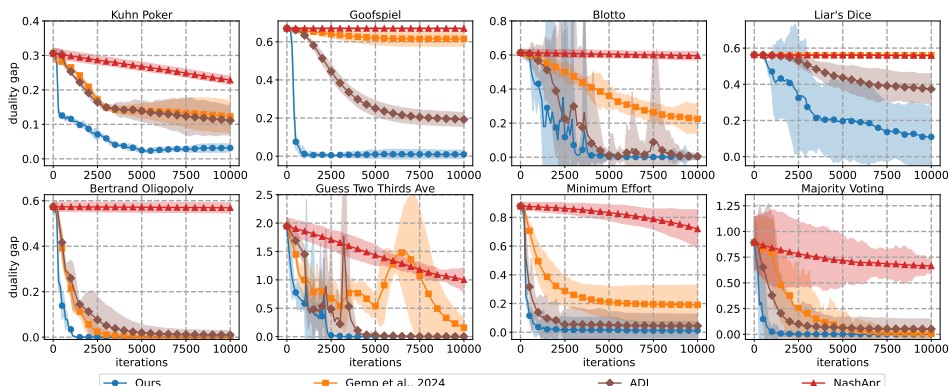

Figure 5: Empirical convergence rates of the tested algorithms when the optimizer is SGD.

**Results on differences between estimated and true loss values.** To determine whether NAL serves as an unbiased loss function, we compare the differences between the estimated and true loss values across various algorithms, as shown in Figure 3. Empirical results confirm that NAL behaves as an unbiased loss function, exhibiting significantly smaller differences between true and estimated values compared to other loss functions. More precisely, the difference between the estimated and true values for NAL is usually two orders of magnitude smaller compared to that of other tested loss functions. As the difference between the true value and the estimated value of NAL is considerably smaller than that of other loss functions, we present a more detailed graph highlighting this difference for NAL in Appendix E.

**Results on convergence rates with different optimizers.** We further assess the robustness of our loss function with different optimizers by evaluating performance using other famous optimizers, such as RMSprop (Bottou, 2010) and SGD (Robbins & Monro, 1951), with the parameter fine tuned in the scenario where Adam is used. Key metrics, such as convergence rate, the variance of estimating the value loss function, and the difference between the estimated value and true value of loss functions, are analyzed. The convergence results for RMSprop and SGD are shown in Figures 4 and 5, respectively. RMSprop shows minimal variation in empirical convergence compared to Adam, likely due to its similarity to Adam. In contrast, all other algorithms, except ours, tend to experience significant performance degradation when using SGD. This is likely due to the considerable difference between SGD and momentum-based optimizers like Adam and RMSprop. The results about variances and differences when using RMSprop or SGD as the optimize are in Appendix E. Consistent with the results using Adam, our algorithm exhits the lowest variance and smallest difference.

**Results on sampling times and convergence rates with different sampling methods.** We also present experimental results for algorithms that employ the sampling method from Gemp et al. (2024) and (Gemp et al., 2022), as described in Appendix E. Specifically, we compare sampling times between the method in Gemp et al. (2024) and (Gemp et al., 2022) with the method in Algorithm 1, and evaluate the convergence rates of algorithms employing the sampling method used in Gemp et al. (2024) and (Gemp et al., 2022) as well as the sampling method in Algorithm 1, respectively. Experimental results show that both the sampling method in Algorithm 1 and our loss function, NAL, significantly enhance the convergence rate.

## 6 CONCLUSIONS

We introduce a novel loss function for using DL to compute an NE, named NAL. This loss function can be estimated without bias, and will incur extremely lower variance than the existing unbiased loss function. In addition, an NE is only a stationary point of NAL rather than having to be a global minimum. Experimental results show that the algorithm minimizing NAL significantly outperforms other tested algorithms.

Our approach offers a promising new direction for computing an NE, with the potential to address the challenges posed by large-scale games. One direction of our future works is to extend our approach to solve imperfect information extensive-form games.

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

## A  DISCUSSION ON LEARNING NASH EQUILIBRIA VIA DEEP LEARNING

Deep learning has emerged as a powerful framework for solving complex optimization and decision-making problems across a wide range of domains. At its core, deep learning leverages neural networks to approximate high-dimensional, non-linear functions, making it particularly suited for modeling strategic interactions in multi-agent systems and solving NE (LeCun et al., 2015; Goodfellow, 2016). Unlike traditional methods, which often rely on explicit mathematical formulations or exhaustive enumeration of strategies, deep learning-based approaches can generalize across diverse scenarios by learning directly from minimizing a loss function (Silver et al., 2017; Mnih et al., 2015).

One of the primary strengths of deep learning is its expressive power, allowing it to capture highly complex patterns and relationships that are difficult to model with traditional methods. Neural networks, especially deep architectures, can approximate arbitrary non-linear functions (Hornik, 1991), making them well-suited for representing the strategic decision-making process in game theory. The ability of deep learning models to learn from raw data—without the need for hand-crafted features—enables them to uncover intricate equilibrium strategies that might otherwise be overlooked (Brown et al., 2019; Goktas et al., 2022; Marris et al., 2022; Liu et al., 2022).

A series of deep learning DL-based methods have been developed to learn NE. To the best of our knowledge, the first work on learning NE via DL is proposed by Duan et al. (2023), which demonstrates the feasibility of using DL to learn NE. Subsequently, Marris et al. (2022); Goktas et al. (2022); Liu et al. (2022) conduct extensive research on network architectures tailored for learning NE through DL. However, these architectures are unsuitable for solving real-world games, as they assume that the payoff matrix can be fully loaded into memory as input to the network. In real-world games, the payoff matrix is often too large to fit into memory, necessitating solutions based on sampling a subset of the matrix, referred to as sampled play.

Under sampled play, designing appropriate loss functions specifically tailored for equilibrium computation remains a significant challenge. Many existing loss functions rely on sampling to estimate gradients or payoffs, which can introduce significant biases (Duan et al., 2023; Gemp et al., 2022) or result in high variance (Gemp et al., 2024), making training unstable. This limitation underscores the need for further research into designing unbiased, low-variance loss functions that are better aligned with the requirements of equilibrium computation. Nonetheless, the intersection of deep learning and game theory offers a promising avenue for solving complex, real-world problems, from economic markets to game theory.

## B  MISSING PROOFS IN SECTION 4

### B.1  PROOF OF LEMMA 4.1

*Proof.* Let $\boldsymbol{b} = (b_1, b_2, \ldots, b_d) \in \mathbb{R}^n$ and $\boldsymbol{y} = (y_1, y_2, \ldots, y_d)$ be a vector in the standard simplex, i.e., $y_i \geq 0$ and $\sum_{k=1}^{n} y_k = 1$.

The inner product $\langle \boldsymbol{b}, \boldsymbol{y} \rangle$ is defined as

$$\langle \boldsymbol{b}, \boldsymbol{y} \rangle = \sum_{k=1}^{n} b_k y_k.$$

We will now prove the lemma in two parts: sufficiency and necessity.

**Sufficiency:**

Assume that all coordinates of $\boldsymbol{b}$ are equal, i.e., $b_1 = b_2 = \cdots = b_d = c$, where $c$ is some constant. In this case, we have

$$\langle \boldsymbol{b}, \boldsymbol{y} \rangle = \sum_{k=1}^{n} b_k y_k = \sum_{k=1}^{n} c y_k = c \sum_{k=1}^{n} y_k = c \cdot 1 = c.$$

Thus,

$$\boldsymbol{b} - \langle \boldsymbol{b}, \boldsymbol{y} \rangle \mathbf{1} = (c, c, \ldots, c) - c = (0, 0, \ldots, 0),$$

which implies that $\boldsymbol{b} - \langle \boldsymbol{b}, \boldsymbol{y} \rangle \mathbf{1} = \mathbf{0}$. Hence, the sufficiency holds.

**Necessity:**

Now, assume that $\boldsymbol{b} - \langle \boldsymbol{b}, \boldsymbol{y}\rangle \mathbf{1} = \mathbf{0}$. We need to show that this implies that all coordinates of $\boldsymbol{b}$ are equal. From the equation $\boldsymbol{b} - \langle \boldsymbol{b}, \boldsymbol{y}\rangle \mathbf{1} = \mathbf{0}$, we have

$$\boldsymbol{b} = \langle \boldsymbol{b}, \boldsymbol{y}\rangle \mathbf{1},$$

where $\mathbf{1} = (1, 1, \dots, 1)$ is the vector of all ones. This implies that

$$b_k = \langle \boldsymbol{b}, \boldsymbol{y}\rangle \quad \text{for all } k = 1, 2, \dots, d.$$

In other words, all $b_k$ are equal to $\langle \boldsymbol{b}, \boldsymbol{y}\rangle$, meaning $b_1 = b_2 = \cdots = b_d$. Thus, the necessity holds.

Since both sufficiency and necessity have been proven, the lemma is true. $\qquad\square$

### B.2    PROOF OF THEOREM 4.2

*Proof.* We prove Theorem 4.2 via the tangent residual (Cai et al., 2022). Therefore, before we start the proof, we first introduce tangent residual. Formally, for any game, whose the utility function $u(\cdot)$ of each player $i$ is concave over $\boldsymbol{\mathcal{X}}_i, \forall \boldsymbol{x} \in \boldsymbol{\mathcal{X}}$, its tangent residual is

$$r^{tan}(\boldsymbol{x}) = \min_{\boldsymbol{z} \in \mathcal{N}_{\boldsymbol{\mathcal{X}}}(\boldsymbol{x})} \| -\nabla_{\boldsymbol{x}} u(\boldsymbol{x}) + \boldsymbol{z}\|_2,$$

where $\mathcal{N}_{\boldsymbol{\mathcal{X}}}(\boldsymbol{x}) = \{\boldsymbol{v} \in \mathbb{R}^{|\boldsymbol{\mathcal{X}}|} : \langle \boldsymbol{v}, \boldsymbol{x}' - \boldsymbol{x}\rangle \le 0, \forall \boldsymbol{x}' \in \boldsymbol{\mathcal{X}}\}$ is the normal cone of $\boldsymbol{x}$, and $\nabla_{\boldsymbol{x}} u(\boldsymbol{x}) = [\nabla_{\boldsymbol{x}_0} u_0(\boldsymbol{x}); \nabla_{\boldsymbol{x}_1} u_1(\boldsymbol{x}); \cdots; \nabla_{\boldsymbol{x}_{n-1}} u_{n-1}(\boldsymbol{x})]$. If $r^{tan}(\boldsymbol{x}) = 0$, then by Lemma B.1, $\boldsymbol{x}$ is an NE. Additionally, if $\boldsymbol{x}$ is an NE, then $\nabla_{\boldsymbol{x}} u(\boldsymbol{x}) \in \mathcal{N}_{\boldsymbol{\mathcal{X}}}(\boldsymbol{x})$, since $\langle \nabla_{\boldsymbol{x}} u(\boldsymbol{x}), \boldsymbol{x}' - \boldsymbol{x}\rangle \le 0, \forall \boldsymbol{x}' \in \boldsymbol{\mathcal{X}}$ when $\boldsymbol{x}$ is an NE. Therefore, $r^{tan}(\boldsymbol{x}) = \| -\nabla_{\boldsymbol{x}} u(\boldsymbol{x}) + \nabla_{\boldsymbol{x}} u(\boldsymbol{x})\|_2 = 0$ if $\boldsymbol{x}$ is an NE.

**Lemma B.1.** *(Proof is in Appendix B.3) For any game, whose utility function of each player $i$ is concave over $\boldsymbol{\mathcal{X}}_i$, it holds that*

$$DualityGap(\boldsymbol{x}) \le C_0 r^{tan}(\boldsymbol{x}),$$

*where $C_0$ is a game-dependent constant.*

From the definition of $-\langle \boldsymbol{F}_i^{\tau, \boldsymbol{x}}, \boldsymbol{x}'_i\rangle \mathbf{1}, \forall \boldsymbol{x}, \boldsymbol{x}' \in \boldsymbol{\mathcal{X}}$, we have

$$\sum_{i \in \mathcal{N}} \langle -\langle \boldsymbol{F}_i^{\tau, \boldsymbol{x}}, \hat{\boldsymbol{x}}_i\rangle \mathbf{1}, \boldsymbol{x}'_i - \boldsymbol{x}_i\rangle = \sum_{i \in \mathcal{N}} -\langle \boldsymbol{F}_i^{\tau, \boldsymbol{x}}, \hat{\boldsymbol{x}}_i\rangle + \langle \boldsymbol{F}_i^{\tau, \boldsymbol{x}}, \hat{\boldsymbol{x}}_i\rangle = 0,$$

where the first equality comes from $\langle \mathbf{1}, \boldsymbol{x}'_i\rangle = \langle \mathbf{1}, \boldsymbol{x}_i\rangle = 1$ since $\boldsymbol{x}'_i$ and $\boldsymbol{x}_i$ are in the simplex. Therefore, $[-\langle \boldsymbol{F}_i^{\tau, \boldsymbol{x}}, \hat{\boldsymbol{x}}_0\rangle \mathbf{1}, -\langle \boldsymbol{F}_i^{\tau, \boldsymbol{x}}; \hat{\boldsymbol{x}}_1\rangle \mathbf{1}; \cdots; -\langle \boldsymbol{F}_i^{\tau, \boldsymbol{x}}, \hat{\boldsymbol{x}}_{|\mathcal{N}|-1}\rangle \mathbf{1}]$ is in the normal cone of $\boldsymbol{x}$. Then, from the definition of the tangent residual,

$$r^{\tan, \tau}(\boldsymbol{x}) = \min_{\boldsymbol{z} \in \mathcal{N}_{\boldsymbol{\mathcal{X}}}(\boldsymbol{x})} \| -\nabla_{\boldsymbol{x}} u^{\tau}(\boldsymbol{x}) + \boldsymbol{z}\|_2,$$

with $\mathcal{N}_{\boldsymbol{\mathcal{X}}}(\boldsymbol{x})$ is the normal cone of $\boldsymbol{x}$, and

$$-\nabla_{\boldsymbol{x}_i} u_i^{\tau}(\boldsymbol{x}) = \boldsymbol{F}_i^{\tau, \boldsymbol{x}},$$

we have that

$$r^{\tan, \tau}(\boldsymbol{x}) \le \|\nabla_{\boldsymbol{x}} \mathcal{L}_{NAL}^{\tau}(\boldsymbol{x})\|_2.$$

In addition, from Lemma B.1, we can obtain that

$$DualityGap^{\tau}(\boldsymbol{x}) \le C_0 r^{\tan, \tau}(\boldsymbol{x}) \le C_0 \|\nabla_{\boldsymbol{x}} \mathcal{L}_{NAL}^{\tau}(\boldsymbol{x})\|_2.$$

It completes the proof. $\qquad\square$

### B.3   PROOF OF LEMMA B.1

*Proof.* Let $\boldsymbol{x}_i' = \arg\max_{\boldsymbol{x}_i' \in \boldsymbol{\mathcal{X}}_i} \langle \nabla_{\boldsymbol{x}_i} u_i(\boldsymbol{x}), \boldsymbol{x}_i' - \boldsymbol{x}_i \rangle$, for the definition of the duality gap and normal cone, $\forall \boldsymbol{z} \in \mathcal{N}_{\boldsymbol{\mathcal{X}}}(\boldsymbol{x})$, we have

$$\begin{aligned} \text{DualityGap}(\boldsymbol{x}) &= \sum_{i \in \mathcal{N}} \langle \nabla_{\boldsymbol{x}_i} u_i(\boldsymbol{x}), \boldsymbol{x}_i' - \boldsymbol{x}' \rangle \\ &\leq \sum_{i \in \mathcal{N}} \langle \nabla_{\boldsymbol{x}_i} u_i(\boldsymbol{x}), \boldsymbol{x}_i' - \boldsymbol{x}_i \rangle + \langle \boldsymbol{z}, \boldsymbol{x} - \boldsymbol{x}' \rangle \\ &= \langle -\nabla_{\boldsymbol{x}} u(\boldsymbol{x}) + \boldsymbol{z}, \boldsymbol{x} - \boldsymbol{x}' \rangle \\ &\leq \| -\nabla_{\boldsymbol{x}} u(\boldsymbol{x}) + \boldsymbol{z} \|_2 \| \boldsymbol{x} - \boldsymbol{x}' \|_2, \end{aligned} \tag{7}$$

where the second lines comes from the fact that, $\forall \boldsymbol{z} \in \mathcal{N}_{\boldsymbol{\mathcal{X}}}(\boldsymbol{x})$ and $\boldsymbol{x}'' \in \boldsymbol{\mathcal{X}}, \langle \boldsymbol{z}, \boldsymbol{x} - \boldsymbol{x}'' \rangle \geq 0$ holds. As Eq. (7) holds for all $\boldsymbol{z} \in \mathcal{N}_{\boldsymbol{\mathcal{X}}}(\boldsymbol{x})$, we can get

$$\text{DualityGap}(\boldsymbol{x}) \leq \| \boldsymbol{x} - \boldsymbol{x}' \|_2 \min_{\boldsymbol{z} \in \mathcal{N}_{\boldsymbol{\mathcal{X}}}(\boldsymbol{x})} \| -\nabla_{\boldsymbol{x}} u(\boldsymbol{x}) + \boldsymbol{z} \|_2,$$

which implies

$$\text{DualityGap}(\boldsymbol{x}) \leq C_0 r^{\tan}(\boldsymbol{x}),$$

where $C_0 = \max_{\boldsymbol{x}'', \boldsymbol{x}''' \in \mathcal{X}} \| \boldsymbol{x}'' - \boldsymbol{x}''' \|_2$. It completes the proof. $\qquad\square$

### B.4   PROOF OF THEOREM 4.3

*Proof.* Beginning with the definition of the duality gap, we find

$$\begin{aligned} &\text{DualityGap}(\boldsymbol{x}) \\ &= \sum_{i \in \mathcal{N}} \max_{\boldsymbol{x}_i' \in \boldsymbol{\mathcal{X}}_i} \langle \nabla_{\boldsymbol{x}_i} u_i(\boldsymbol{x}), \boldsymbol{x}_i' - \boldsymbol{x}_i \rangle \\ &= \sum_{i \in \mathcal{N}} \max_{\boldsymbol{x}_i' \in \boldsymbol{\mathcal{X}}_i} \langle \nabla_{\boldsymbol{x}_i} u_i(\boldsymbol{x}) - \tau \log(\boldsymbol{x}_i) + \tau \log(\boldsymbol{x}_i), \boldsymbol{x}_i' - \boldsymbol{x}_i \rangle \\ &\leq \sum_{i \in \mathcal{N}} \max_{\boldsymbol{x}_i' \in \boldsymbol{\mathcal{X}}_i} \langle \nabla_{\boldsymbol{x}_i} u_i(\boldsymbol{x}) - \tau \log(\boldsymbol{x}_i), \boldsymbol{x}_i' - \boldsymbol{x}_i \rangle + \sum_{i \in \mathcal{N}} \max_{\boldsymbol{x}_i' \in \boldsymbol{\mathcal{X}}_i} \langle \tau \log(\boldsymbol{x}_i), \boldsymbol{x}_i' - \boldsymbol{x}_i \rangle \\ &\leq \text{DualityGap}^\tau(\boldsymbol{x}) + \sum_{i \in \mathcal{N}} \max_{\boldsymbol{x}_i' \in \boldsymbol{\mathcal{X}}_i} \langle \tau \log(\boldsymbol{x}_i), \boldsymbol{x}_i' \rangle + \sum_{i \in \mathcal{N}} \langle \tau \log(\boldsymbol{x}_i), -\boldsymbol{x}_i \rangle \\ &\leq C_0 \| \nabla_{\boldsymbol{x}} \mathcal{L}_{NAL}^\tau(\boldsymbol{x}) \|_2 + \sum_{i \in \mathcal{N}} \langle \tau \log(\boldsymbol{x}_i), -\boldsymbol{x}_i \rangle \\ &\leq C_0 \| \nabla_{\boldsymbol{x}} \mathcal{L}_{NAL}^\tau(\boldsymbol{x}) \|_2 + \tau \sum_{i \in \mathcal{N}} \log(|\mathcal{A}_i|). \end{aligned}$$

where the second inequality follows from the definition of the duality gap and NE, and the third inequality comes from $\log(x_i) \leq 0, \forall 0 \geq x_i \leq 1$. It completes the proof. $\qquad\square$

## C   VARIANCE OF ESTIMATING VIA TWO INDEPENDENT AND IDENTICALLY DISTRIBUTED RANDOM VARIABLES

Let two independent samples from two corresponding identically distributed are $Y^{(1)}$ and $Y^{(2)}$. Due to the definition of $Y^{(1)}$ and $Y^{(2)}$, we have $\mathbb{E}[Y^{(1)}] = \mathbb{E}[Y^{(2)}] = Y$. Assume $\text{Var}[Y^{(1)}] \leq \sigma$ and $\text{Var}[Y^{(2)}] \leq \sigma$. Now, we aim to analyze the variance $\text{Var}[Y^{(1)} Y^{(2)}]$. From the definition of $\text{Var}[Y^{(1)} Y^{(2)}]$, we have

$$\text{Var}[Y^{(1)} Y^{(2)}] = \mathbb{E}[(Y^{(1)})^2] \mathbb{E}[(Y^{(2)})^2] - (\mathbb{E}[Y^{(1)}] \mathbb{E}[Y^{(2)}])^2.$$

For the term $\mathbb{E}[Y^{(1)}]^2$, from the term $\text{Var}[Y^{(1)}]$, we get

$$\begin{aligned} \text{Var}[Y^{(1)}] &= \mathbb{E}[(Y^{(1)})^2] - (\mathbb{E}[Y^{(1)}])^2 = \mathbb{E}[(Y^{(1)})^2] - Y^2 \leq \sigma \\ &\Leftrightarrow \mathbb{E}[(Y^{(1)})^2] \leq \sigma + Y^2. \end{aligned}$$

Similarly, we have $\mathbb{E}[(Y^{(2)})^2] \le \sigma + Y^2$. Combining the above equities, we have

$$\text{Var}[Y^{(1)}Y^{(2)}] \le (\sigma + Y^2)(\sigma + Y^2) - (Y^2)^2 \le \sigma^2 + 2\sigma Y^2.$$

# D   IMPLEMENTATION DETAILS OF COMPARED LOSS FUNCTIONS

**The loss function proposed by Gemp et al. (2024).**    As shown in Algorithm 1, we do not employ the sampling method used in Gemp et al. (2022) and Gemp et al. (2024) due to the high per-sample sampling complexity of their sampling method. Formally, the per-sample sampling complexity of their estimating method is $O(|\mathcal{N}||\mathcal{A}_i|^2)$ while that of our estimating method as shown in Algorithm 1 is $O(|\mathcal{N}|)$. In Appendix E, we provide a comparison of the sampling time between our sampling method with the sampling method used in Gemp et al. (2022) and Gemp et al. (2024), under the same number of sampled instances $S$. In our sampling method, the estimated variable of $\boldsymbol{F}_i^{\boldsymbol{x}}$ cannot participate in gradient backpropagation, and only the variable $\boldsymbol{x}_i^{\boldsymbol{\theta}}$ participate in gradient backpropagation. In other words, we minimize the following loss function

$$\mathcal{L}_{Gemp}^{\tau}(\boldsymbol{\theta}) = \sum_{i \in \mathcal{N}} \| sg[\boldsymbol{F}_i^{\boldsymbol{x}^{\boldsymbol{\theta}}}] + \tau \log \boldsymbol{x}_i^{\boldsymbol{\theta}} - sg[\overline{\boldsymbol{F}_i^{\boldsymbol{x}^{\boldsymbol{\theta}}}}] + \tau \overline{\log \boldsymbol{x}_i^{\boldsymbol{\theta}}} \|_2^2.$$

As did in Gemp et al. (2024), we use the following loss function to estimate the value of $\mathcal{L}_{Gemp}^{\tau}(\boldsymbol{\theta})$ via the $(2s-1)$-th and $(2s)$-th tuples ($[i, a_i^{2s-1}, r_i^{2s-1}, p_i^{2s-1}]$ and $[i, a_i^{2s}, r_i^{2s}, p_i^{2s}]$) stored in $\mathcal{M}_i$:

$$\tilde{\mathcal{L}}_{Gemp}^{\tau}(\boldsymbol{\theta}) = \sum_{i \in \mathcal{N}} \sum_{s=1}^{s=\frac{S}{2}} \langle sg[\hat{\boldsymbol{F}}_{i,2s-1}^{\boldsymbol{x}^{\boldsymbol{\theta}}}] + \tau \log \boldsymbol{x}_i^{\boldsymbol{\theta}} - sg[\overline{\hat{\boldsymbol{F}}_{i,2s-1}^{\boldsymbol{x}^{\boldsymbol{\theta}}}}] + \tau \overline{\log \boldsymbol{x}_i^{\boldsymbol{\theta}}}, sg[\hat{\boldsymbol{F}}_{i,2s}^{\boldsymbol{x}^{\boldsymbol{\theta}}}] + \tau \log \boldsymbol{x}_i^{\boldsymbol{\theta}} - sg[\overline{\hat{\boldsymbol{F}}_{i,2s}^{\boldsymbol{x}^{\boldsymbol{\theta}}}}] + \tau \overline{\log \boldsymbol{x}_i^{\boldsymbol{\theta}}} \rangle,$$

$$\hat{\boldsymbol{F}}_{i,2s-1}^{\boldsymbol{x}^{\boldsymbol{\theta}}} = \frac{r_i^{2s-1} - \tau \log p_i^{2s-1}}{p_i^{2s-1}} \boldsymbol{e}_{a_i^{2s-1}}, \quad \hat{\boldsymbol{F}}_{i,2s}^{\boldsymbol{x}^{\boldsymbol{\theta}}} = \frac{r_i^{2s} - \tau \log p_i^{2s}}{p_i^{2s}} \boldsymbol{e}_{a_i^{2s}},$$

where $\boldsymbol{e}_{a_i^{2s-1}}$ ($\boldsymbol{e}_{a_i^{2s}}$) is a vector in which the coordinate $a_i^{2s-1}$ ($a_i^{2s}$) is 1 and all other coordinates are 0. From the analysis in Gemp et al. (2024), $\mathbb{E}[\hat{\boldsymbol{F}}_{i,2s-1}^{\boldsymbol{x}^{\boldsymbol{\theta}}}] = \boldsymbol{F}_i^{\boldsymbol{x}^{\boldsymbol{\theta}}}$, $\mathbb{E}[\hat{\boldsymbol{F}}_{i,2s}^{\boldsymbol{x}^{\boldsymbol{\theta}}}] = \boldsymbol{F}_i^{\boldsymbol{x}^{\boldsymbol{\theta}}}$, and $\frac{2}{S}\mathbb{E}[\tilde{\mathcal{L}}_{Gemp}^{\tau}(\boldsymbol{\theta})] = \hat{\mathcal{L}}_{Gemp}^{\tau}(\boldsymbol{\theta})$.

**ADI (Gemp et al., 2022).**    Since the variable $\boldsymbol{F}_i^{\boldsymbol{x}}$ cannot participate in gradient backpropagation via our sampling method, we defined this loss function as

$$\mathcal{L}_{ADI}^{\tau}(\boldsymbol{\theta}) = \sum_{i \in \mathcal{N}} \max_{\boldsymbol{x}_i' \mathcal{X}_i} \langle sg[\boldsymbol{F}_i^{\boldsymbol{x}^{\boldsymbol{\theta}}}] + \tau \log \boldsymbol{x}_i^{\boldsymbol{\theta}}, \boldsymbol{x}' - \boldsymbol{x}_i^{\boldsymbol{\theta}} \rangle.$$

We use the following loss function to estimate the value of $\mathcal{L}_{ADI}^{\tau}(\boldsymbol{\theta})$ via the the tuples $[i, a_i^s, r_i^s, p_i^s]$ ($s \in [1, 2, \cdots, S]$) stored in $\mathcal{M}_i$:

$$\hat{\boldsymbol{F}}_{i,s}^{\boldsymbol{x}^{\boldsymbol{\theta}}} = \frac{r_i^s - \tau \log p_i^s}{p_i^s} \boldsymbol{e}_{a_i^s}, \quad \hat{\boldsymbol{F}}_i^{\boldsymbol{x}^{\boldsymbol{\theta}}} = \sum_{s=1}^{s=S} \hat{\boldsymbol{F}}_{i,s}^{\boldsymbol{x}^{\boldsymbol{\theta}}},$$

$$\tilde{\mathcal{L}}_{ADI}^{\tau}(\boldsymbol{\theta}) = \sum_{i \in \mathcal{N}} \max_{\boldsymbol{x}_i' \mathcal{X}_i} \langle sg[\hat{\boldsymbol{F}}_i^{\boldsymbol{x}^{\boldsymbol{\theta}}}] + \tau \log \boldsymbol{x}_i^{\boldsymbol{\theta}}, \boldsymbol{x}' - \boldsymbol{x}_i^{\boldsymbol{\theta}} \rangle.$$

**NashApr (Duan et al., 2023).**    Since the variable $\boldsymbol{F}_i^{\boldsymbol{x}}$ cannot participate in gradient backpropagation via our sampling method, we defined this loss function as

$$\mathcal{L}_{NashApr}(\boldsymbol{\theta}) = \max_{i \in \mathcal{N}} \max_{\boldsymbol{x}_i' \mathcal{X}_i} \langle sg[\boldsymbol{F}_i^{\boldsymbol{x}^{\boldsymbol{\theta}}}], \boldsymbol{x}' - \boldsymbol{x}_i^{\boldsymbol{\theta}} \rangle.$$

We use the following loss function to estimate the value of $\mathcal{L}_{NashApr}(\boldsymbol{\theta})$ via the the tuples $[i, a_i^s, r_i^s, p_i^s]$ ($s \in [1, 2, \cdots, S]$) stored in $\mathcal{M}_i$:

$$\hat{\boldsymbol{F}}_{i,s}^{\boldsymbol{x}^{\boldsymbol{\theta}}} = \frac{r_i^s - \tau \log p_i^s}{p_i^s} \boldsymbol{e}_{a_i^s}, \quad \hat{\boldsymbol{F}}_i^{\boldsymbol{x}^{\boldsymbol{\theta}}} = \sum_{s=1}^{s=S} \hat{\boldsymbol{F}}_{i,s}^{\boldsymbol{x}^{\boldsymbol{\theta}}}, \quad \tilde{\mathcal{L}}_{NashApr}^{\tau}(\boldsymbol{\theta}) = \max_{i \in \mathcal{N}} \max_{\boldsymbol{x}_i' \mathcal{X}_i} \langle sg[\hat{\boldsymbol{F}}_i^{\boldsymbol{x}^{\boldsymbol{\theta}}}], \boldsymbol{x}' - \boldsymbol{x}_i^{\boldsymbol{\theta}} \rangle.$$

# E  ADDITIONAL EXPERIMENTAL RESULTS

**Results on differences between true and estimated values for NAL.** Firstly, as previously mentioned, the difference between true and estimated values for NAL is significantly smaller compared to other loss functions. To illustrate this difference more clearly, we present a detailed graph specifically for NAL, as shown in Figure 6.

**Results on variances and differences between true and estimated values with different optimizers.** Secondly, we present the results of the variance in estimating the value of loss functions and the difference between the true and estimated values when RMSprop and SGD are used as optimizers. The variance in estimating the value of loss functions under RMSprop and SGD are shown in Figures 7 and 8, respectively. Similar to the case when Adam is used as the optimizer, our algorithm demonstrates the lowest variance. Additionally, the difference between the true and estimated loss values under RMSprop and SGD are presented in Figures 9 and 10, respectively. Again, consistent with the results using Adam, our algorithm exhibits the smallest difference.

**Results on convergence rates with different numbers of sampled instances.** Then, we investigate the empirical convergence rates of our algorithm and that of Gemp et al. (2024), with varying numbers of sampled instances $S$ per iteration, using Adam as the optimizer. We focus on these two algorithms since they both minimize unbiased loss functions. The results are presented in Figure 11. We observe that as $S$ increases, our algorithm converge to a more and more accurate NE. In contrast, the algorithm proposed by Gemp et al. (2024) learns a more and more accurate NE in Liar's Dice, but fails to learn a more accurate NE in Kuhn Poker. These results demonstrate that reaching the global minimum of the loss function is challenging, while finding a stationary point is easier.

**Results on sampling times with different sampling methods.** We also provide a comparison of the sampling time between our sampling method in Algorithm 1 with the sampling method used in Gemp et al. (2022) and Gemp et al. (2024), under the same number of sampled instances $S$. We conduct our tests on Liar's Dice because it has the largest number of actions for each player among the eight games evaluated in the experiments. The results are shown in Figure 12. We observe that the sampling time of the methods employed by Gemp et al. (2022) and Gemp et al. (2024) is at least 10,000 times greater than that of our sampling method. Specifically, when $S = 10$, which corresponds to the configuration used in our experiments (Section 5), our sampling method achieves a sampling time of approximately 0.004 seconds, while the methods from Gemp et al. (2022) and Gemp et al. (2024) require about 1590 seconds.

**Results on convergence rates with different sampling methods in terms of time.** Now, we compare the convergence rates of algorithms employing different sampling methods in terms of time. We focus on algorithms that minimize NAL or the loss function proposed by Gemp et al. (2024), the only known unbiased loss functions. We conduct experiments on Liar's Dice, which has the largest number of actions (2306) among all tested games. For the algorithms employing the sampling method outlined in Algorithm 1, we set $S = 100$ to learn a sufficiently accurate approximation of NE, while maintaining all other parameters as described in Section 5. For the algorithms that adopt the sampling method utilized in Gemp et al. (2022) and Gemp et al. (2024), We reduce $T$ and $T_u$ in Section 5 by a factor of 100 since the sampling time associated with the methods in Gemp et al. (2022) and Gemp et al. (2024) is excessively large. In addition, we employ two different settings of the value of $S$, *e.g.*, $S = 2$ and $S = 100$. All other settings remain unchanged from Section 5. Notably, when employing the loss function from Gemp et al. (2024), $\boldsymbol{F}_i^{\boldsymbol{x}^{\boldsymbol{\theta}}}$ also contributes to the gradient backpropagation, consistent with the settings in the original paper by Gemp et al. (2024). The experimental results are presented in Figure 13. We observe that both algorithms minimizing NAL exhibit a faster convergence rate than minimizing the loss function proposed by Gemp et al. (2024). More importantly, we find that the wall times of the algorithms utilizing the sampling methods from Gemp et al. (2022) and Gemp et al. (2024) are significantly greater than those of the algorithms employing the sampling method in Algorithm 1. This suggests that, when addressing real-world games, the sampling method in Algorithm 1 is more advantageous, as the action space in real-world scenarios vastly exceeds the 2306 actions present in Liar's Dice.

**Results on convergence rates of algorithms employing the sampling method used in Gemp et al. (2022) and Gemp et al. (2024) in terms of epochs.** Finally, we compare the convergence rates of the algorithms when they employing the sampling method presented in Gemp et al. (2022) and Gemp et al. (2024) across all games, measured in terms of epochs. It is important to note that, when using

the sampling method from Gemp et al. (2022) and Gemp et al. (2024), the runtime of the algorithms is primarily determined by the sampling time. Therefore, the runtime difference between algorithms with the same number of epochs is negligible, which implies that analyzing the convergence rates in terms of epochs is sufficient to reflect the convergence rates in terms of runtime. As did in Figure 13, we focus on algorithms that minimize NAL and the loss function proposed by Gemp et al. (2024). We reduce $T$ and $T_u$ in Section 5 by a factor of 100, and set $S$ to 2 instead of 10, as the sampling time associated with the methods in Gemp et al. (2022) and Gemp et al. (2024) is excessively large. All other settings remain unchanged from Section 5. The experimental results are presented in Figure 14. In alignment with the findings in Section 5, we observe that the algorithm minimizing NAL exhibits a significantly superior convergence rate compared to the algorithm minimizing the loss function proposed by Gemp et al. (2024). More critically, in numerous games, the latter algorithm fails to learn a sufficiently accurate NE. Conversely, the algorithm minimizing NAL successfully learns a accurate NE in nearly all games, characterized by exploitability approaching zero.

**Results on convergence rates of algorithm minimizing NashApr when the optimizer is SGD with larger learning rates in terms of epochs.** We now present the results when the loss function is NashApr, the optimizer is SGD, and the learning rate is increased by 10 times and 100 times compared to the learning rate of NashApr shown in Appendix F. The experimental results are illustrated in Figure 15 and Figure 16, corresponding to learning rates 10 and 100 times higher, respectively, while keeping the learning rates for algorithms using other loss functions unchanged. We observe that algorithms using NashApr as the loss function still perform poorly.

**Results on the value of NAL.** We present the value curves of NAL during the training process. Adam is employed as the optimizer. The parameter $\tau$ remains constant throughout the training because, as observed, shrinking $\tau$ continuously (as suggested in Algorithm 1) renders it excessively small, diminishing its impact. The experimental results are shown in Figure 17. Note that the absence of biased estimates in Goofspiel is an artifact of the logarithmic scaling of the y-axis, leading to a visual distortion. In most cases, we observe that the values of NAL converge to zero, aligning with the NE of the regularization game. However, it is important to emphasize that since the NE in NAL is merely a stationary point, the values of NAL may either exceed or fall below zero.

**Results on convergence rates of NAL with or without $\langle F_i^{\tau,x}, \hat{x}_i \rangle 1$.** We now investigate the performance of NAL when $\langle F_i^{\tau,x}, \hat{x}_i \rangle 1$ is absent. The case where NAL does not include $\langle F_i^{\tau,x}, \hat{x}_i \rangle 1$ can also be interpreted as $\hat{x}_i = 0$. In this scenario, NE is not a stationary point of NAL since the gradient of NAL $F_i^{\tau,x} - \langle F_i^{\tau,x}, \hat{x}_i \rangle 1$ is not $0$ if $x$ is an NE, which may result in the change of parameters of the neural network according to the chain rule, leading to a shift in the strategy. The experimental results are shown in Figure 19. We demonstrate the performance for different values of $S$ ($S = 2, 10, 100$). Note that the minimum value of $S$ must be 2. If $S = 1$, it is not possible to estimate $F_i^{\tau,x} - \langle F_i^{\tau,x}, \hat{x}_i \rangle 1$ via Algorithm 1, as the estimated value of $F_i^{\tau,x} - \langle F_i^{\tau,x}, \hat{x}_i \rangle 1$ in this case will always be $0$ (since $r_i^s - v_i = 0$). From the experimental results, we observe that using NAL with $\hat{x}_i = 0$ as the loss function consistently performs worse than using the standard NAL, especially as $S$ decreases.

In addition, we find that the softmax function in the final layer of our neural network leads to identical parameter updates under $F_i^{\tau,x} - \langle F_i^{\tau,x}, \hat{x}_i \rangle 1$ and $F_i^{\tau,x}$, due to the normalization operation (i.e., for any $z > 0$, it outputs $z / \text{sum}(z)$, which lies in the simplex). Therefore, we conduct additional experiments. In these experiments, we do not use sampling to avoid the effects of estimating, and replace softmax with sparsemax (Martins & Astudillo, 2016), which guarantees that the final output remains within the simplex without the normalization operation. Specifically, we compute the true values of $F_i^{\tau,x}$ and $F_i^{\tau,x} - \langle F_i^{\tau,x}, \hat{x}_i \rangle 1$, rather than estimating them. The results are shown in Figure 20. We observe that the standard NAL outperforms NAL with $\hat{x}_i = 0$ by a large margin. In fact, we never observe convergence for NAL with $\hat{x}_i = 0$. It suggests that, when the activation function is softmax, the convergence of NAL with $\hat{x}_i = 0$ is likely due to the identical parameter updates under $F_i^{\tau,x} - \langle F_i^{\tau,x}, \hat{x}_i \rangle 1$ and $F_i^{\tau,x}$.

# F The Hyperparameters Used in Experiments

In this section, we show the hyperparameters used in Section 5. The hyperparameters for algorithms that minimize NAL, the loss function defined in Eq, (1), ADI, and NashApr, are shown in Table 1, Table 2, Table 3, and Table 4, respectively.

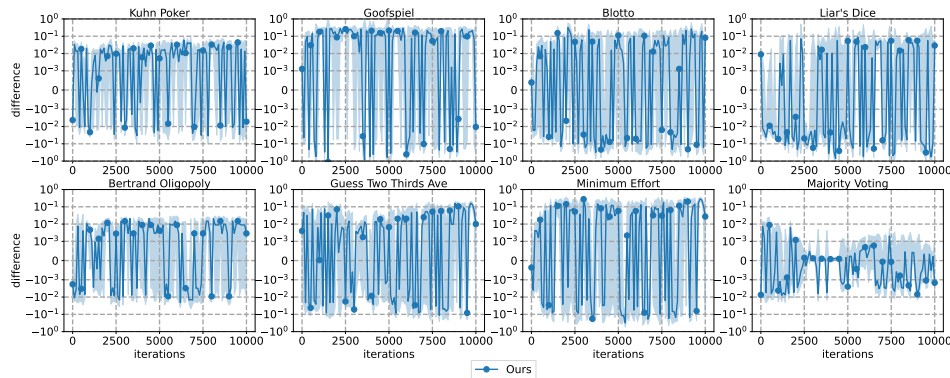

Figure 6: Difference between the true value and the estimated value of NAL when the optimizer is Adam.

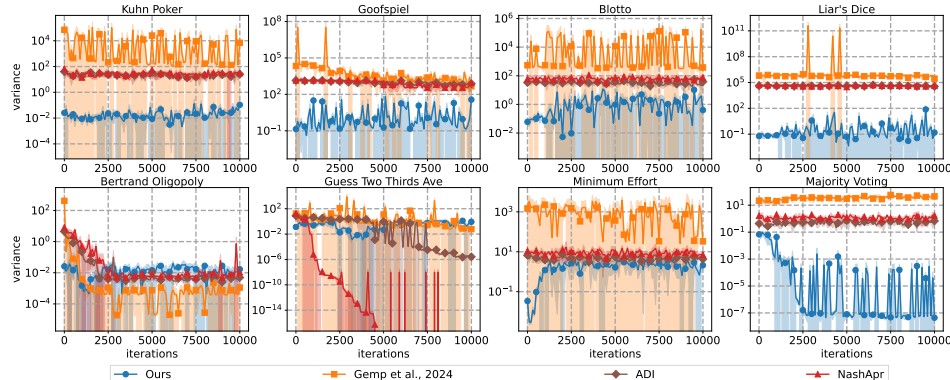

Figure 7: Variances observed in estimating the value of loss functions used by different algorithms when the optimizer is RMSprop.

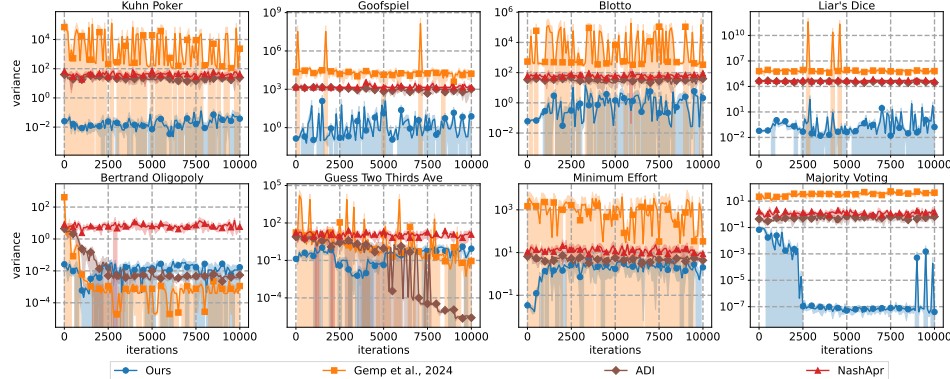

Figure 8: Variances observed in estimating the value of loss functions used by different algorithms when the optimizer is SGD.

Table 1: The hyperparameters of the algorithm that learns an NE via minimizing NAL.

|  | $\eta$ | $\tau$ | $T_u$ | $\alpha$ | $\beta$ |
|---|---|---|---|---|---|
| Kuhn Poker | 0.0001 | 0.1 | 200 | 0.9 | 0.9 |
| Goofspiel | 0.0001 | 0.1 | 200 | 0.9 | 0.5 |
| Blotto | 0.0001 | 0.1 | 500 | 0.9 | 0.5 |
| Liar's Dice | 0.0001 | 0.1 | 500 | 0.9 | 0.5 |
| Bertrand Oligopoly | 0.0001 | 0.1 | 200 | 0.9 | 0.5 |
| Guess Two Thirds Ave | 0.0001 | 0.1 | 1000 | 0.9 | 0.5 |
| Minimum Effort | 0.0001 | 0.1 | 200 | 0.9 | 0.5 |
| Majority Voting | 0.0001 | 0.1 | 200 | 0.9 | 0.5 |

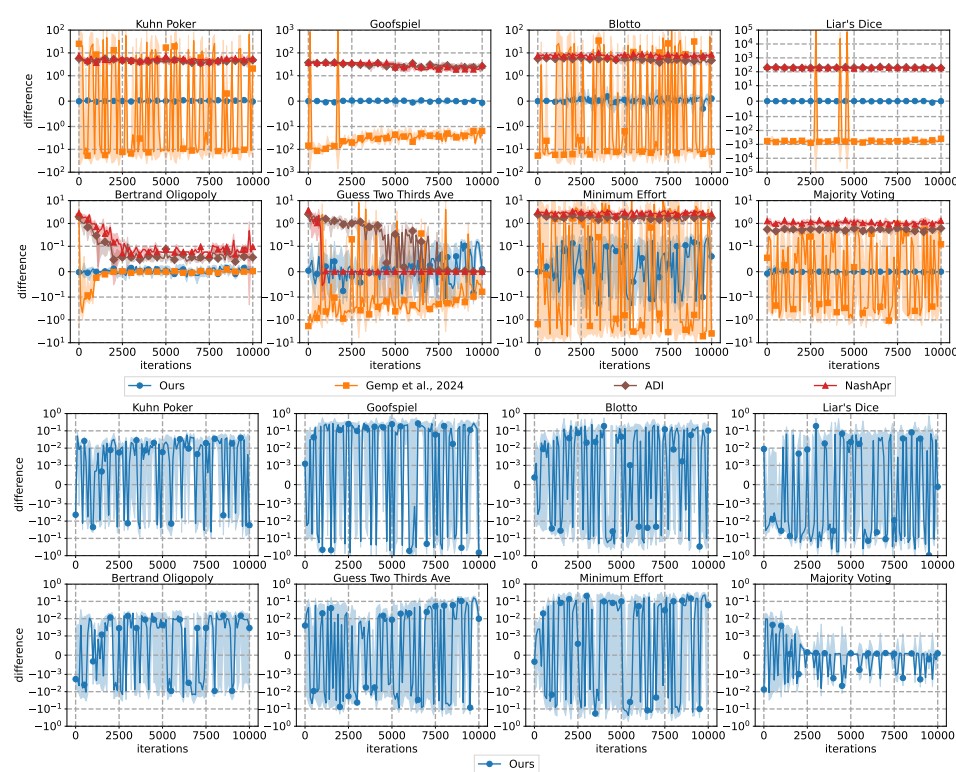

Figure 9: Difference between the true value and the estimated value of loss functions when the optimizer is RMSprop.

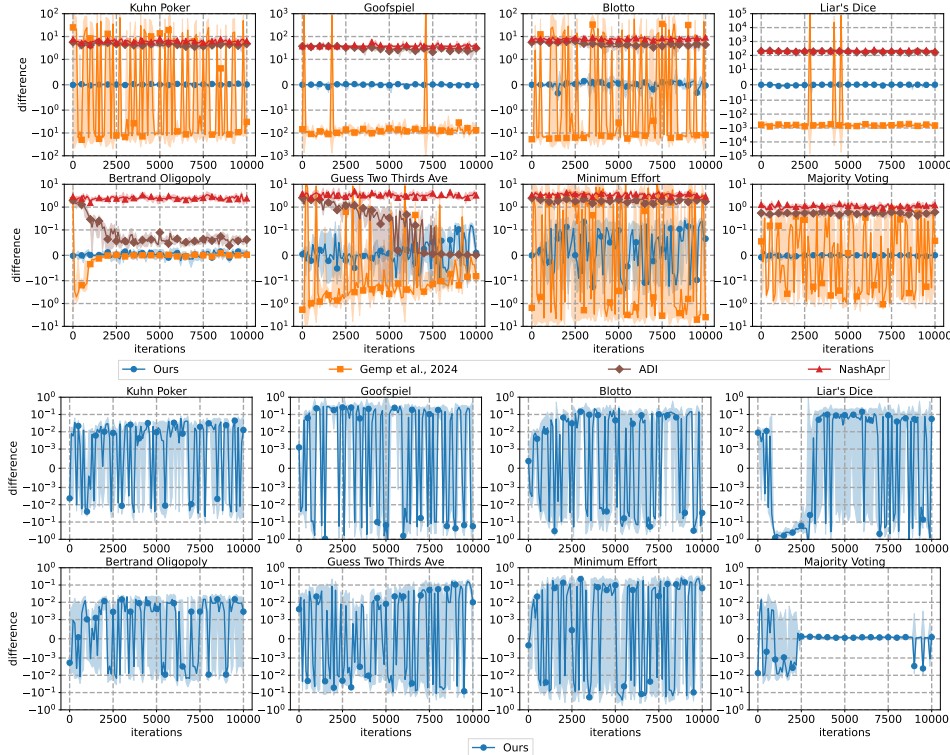

Figure 10: Difference between the true value and the estimated value of loss functions when the optimizer is SGD.

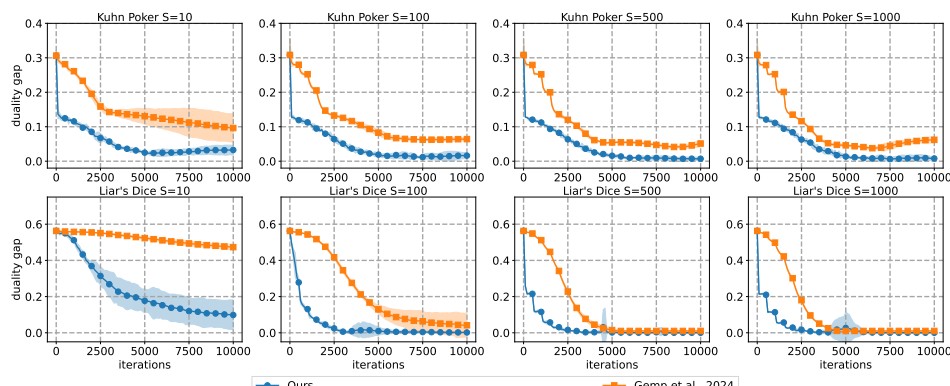

Figure 11: Empirical convergence rates of our algorithm, as well as the algorithm proposed by Gemp et al. (2024), with varying numbers of sampled instances $S$ at per iteration, are evaluated when the optimizer is Adam.

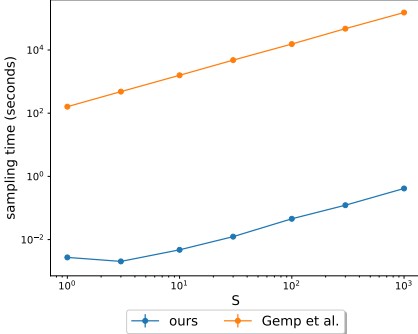

Figure 12: Comparison of the sampling times of our sampling method, shown in Algorithm 1, with the sampling method used in Gemp et al. (2022) and Gemp et al. (2024) for various values of the number $S$ of the sampled instance in Liar's Dice. We conduct our tests on Liar's Dice because it has the largest number of actions for each player among the eight games evaluated in the experiments. For each $S$, we run four seeds and report the average sampling times.

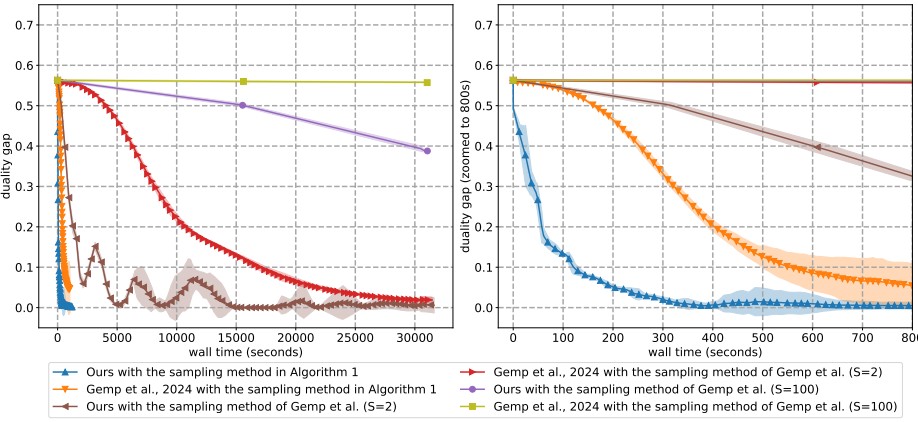

Figure 13: Empirical convergence rates of the algorithms utilizing various sampling methods in Liar's Dice. The x-axis represents the wall time. For algorithms that employ the sampling method outlined in Algorithm 1, the parameter $S$ is set to 100 to ensure the learning of a sufficiently accurate NE. For algorithms that employ the sampling method used in Gemp et al. (2022) and Gemp et al. (2024), we reduce the $T$ and $T_u$ in Section 5 by a factor of 100, as the sampling method used in Gemp et al. (2022) and Gemp et al. (2024) results in excessively higher sampling times for each instance than that of the sampling method in Algorithm 1, which is used in Section 5. The remaining hyperparameters for each algorithms remain consistent with those used in Section 5. The graph on the right is a scaled version of the one on the left.

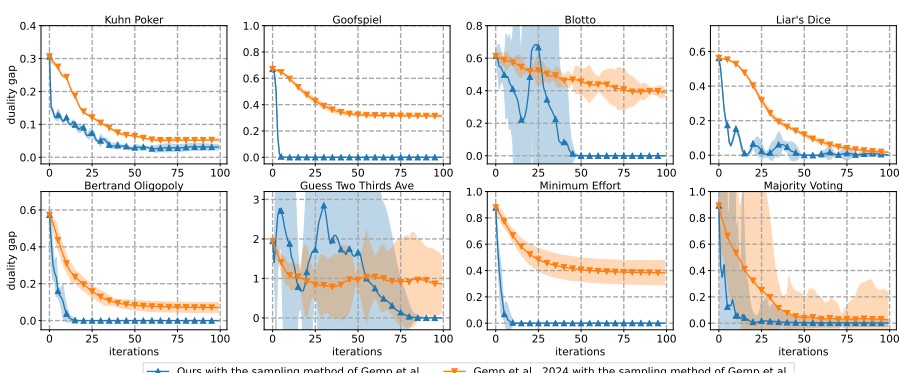

Figure 14: Empirical convergence rates of the algorithms that employ the sampling method used in Gemp et al. (2022) and Gemp et al. (2024). We reduce the $T$ and $T_u$ in Section 5 by a factor of 100, and set $S$ as 2 rather than 10 in Section 5. The remaining hyperparameters remain consistent with those used in Section 5.

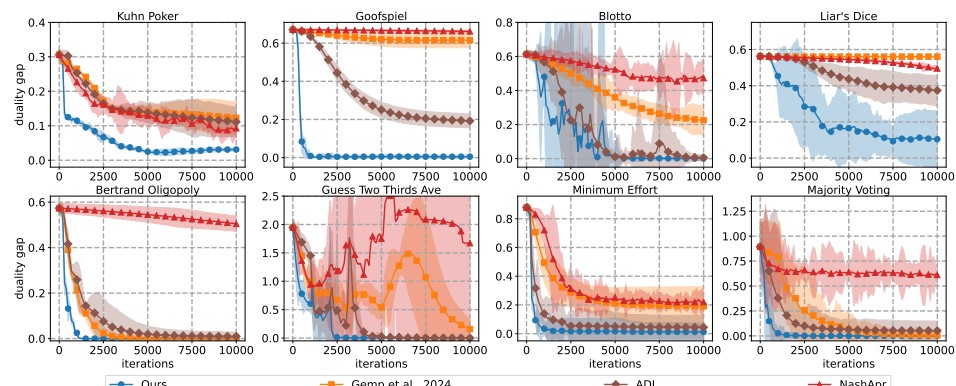

Figure 15: Empirical convergence rates of the tested algorithm when the optimizer is SGD with 10 times larger learning rate for NashApr.

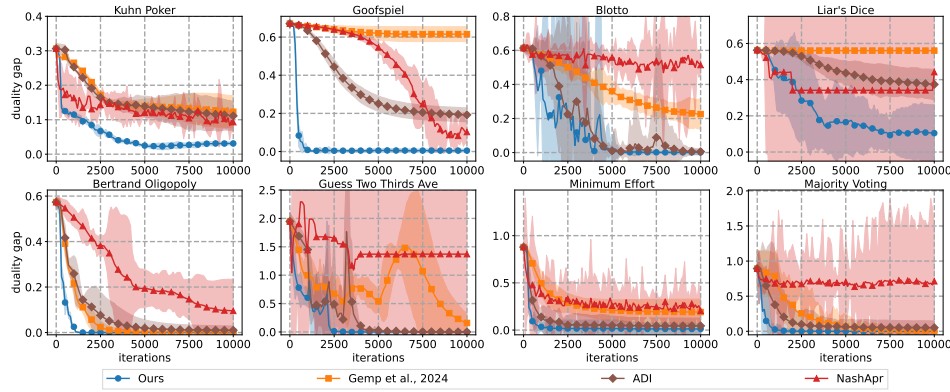

Figure 16: Empirical convergence rates of the tested algorithm when the optimizer is SGD with 100 times larger learning rate for NashApr.

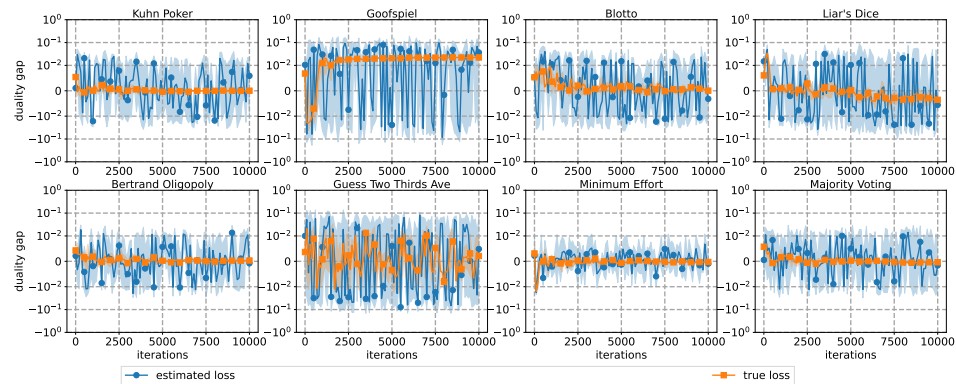

Figure 17: Value curves of NAL during the training. The optimizer is Adam. The parameter $\tau = 0.1$ (from the hyperparameters in Table 1) remains constant. Note that the absence of biased estimates in Goofspiel is an artifact of the logarithmic scaling of the y-axis, leading to a visual distortion.

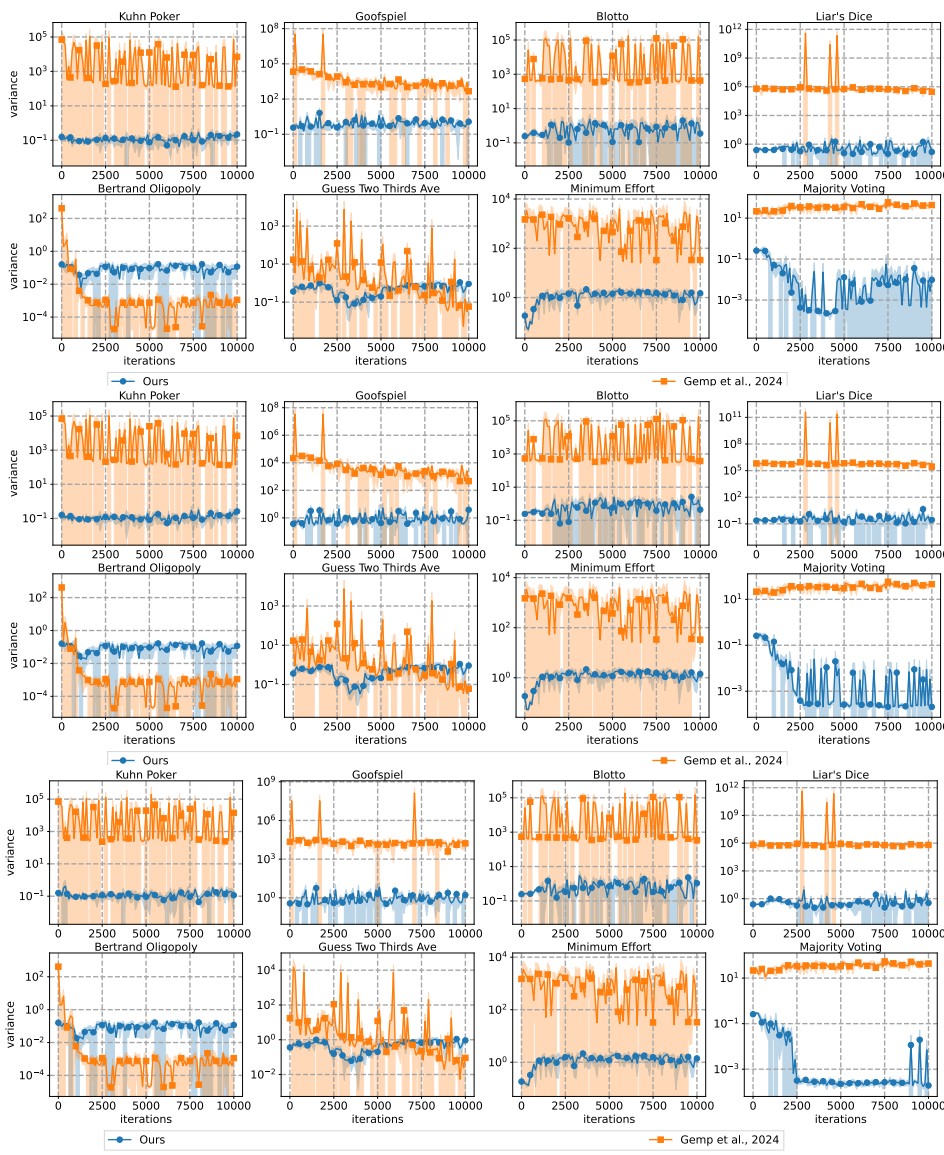

Figure 18: Comparison of the variance of NAL with the square root of the variance of the loss function in Gemp et al. (2024) when the optimizers are Adam (top), RMSprop (middle), and SGD (bottom), respectively. Notably, this figure is only for rebuttal.

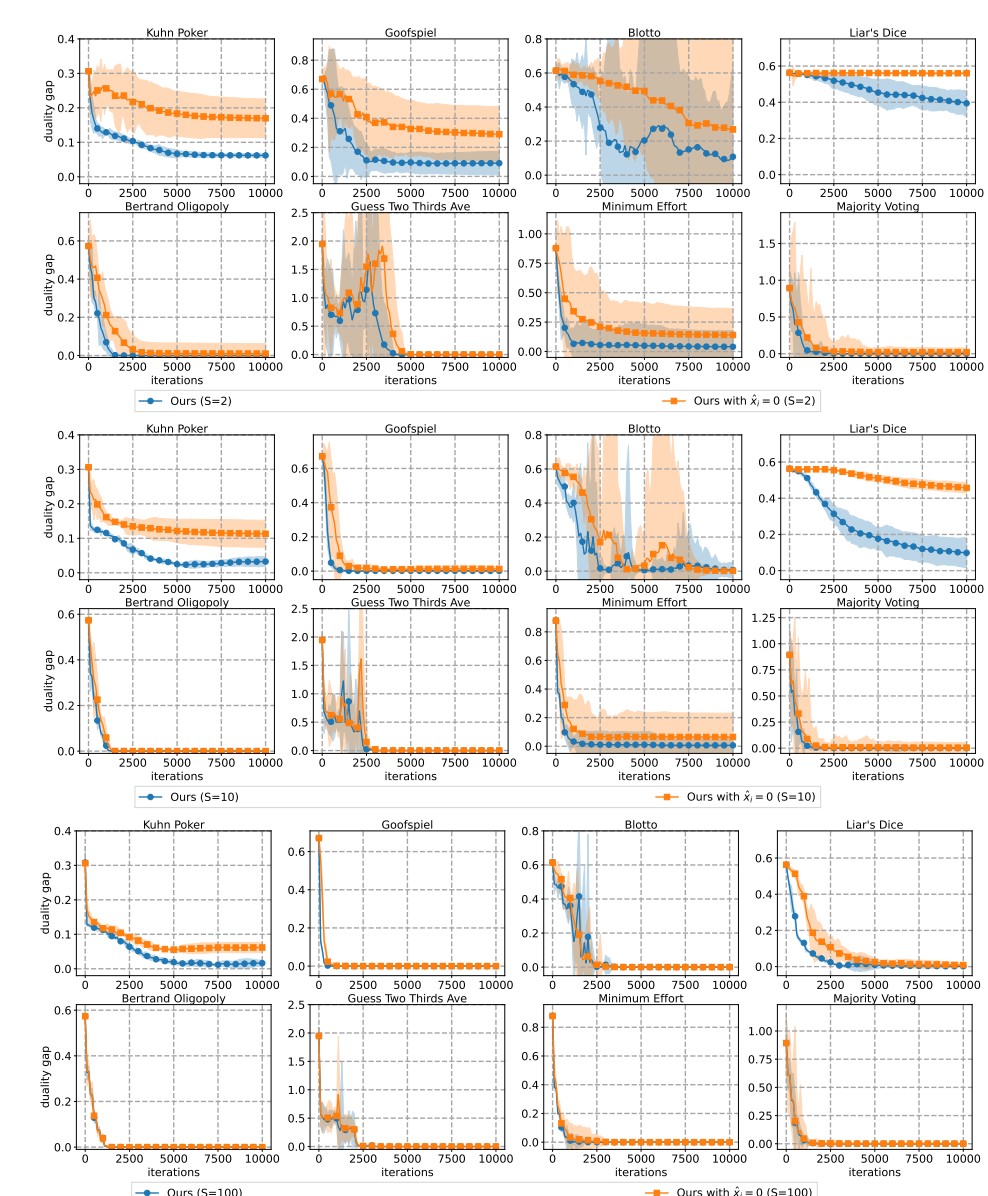

Figure 19: Empirical convergence rates of the algorithms that minimize NAL and NAL with $\hat{x}_i = 0$, respectively, when the optimizer is Adam.

Table 2: The hyperparameters of the algorithm that learns an NE via minimizing the loss function proposed by Gemp et al. (2024).

|  | $\eta$ | $\tau$ | $T_u$ | $\alpha$ | $\beta$ |
|---|---|---|---|---|---|
| Kuhn Poker | 0.00001 | 1 | 500 | 0.9 | 0.5 |
| Goofspiel | 0.00001 | 0.1 | 200 | 0.9 | 0.5 |
| Blotto | 0.00001 | 0.1 | 500 | 0.9 | 0.5 |
| Liar's Dice | 0.00001 | 0.1 | 500 | 0.9 | 0.5 |
| Bertrand Oligopoly | 0.00001 | 0.1 | 200 | 0.9 | 0.5 |
| Guess Two Thirds Ave | 0.00001 | 0.1 | 500 | 0.9 | 0.9 |
| Minimum Effort | 0.00001 | 0.1 | 200 | 0.9 | 0.9 |
| Majority Voting | 0.00001 | 0.1 | 1000 | 0.9 | 0.5 |

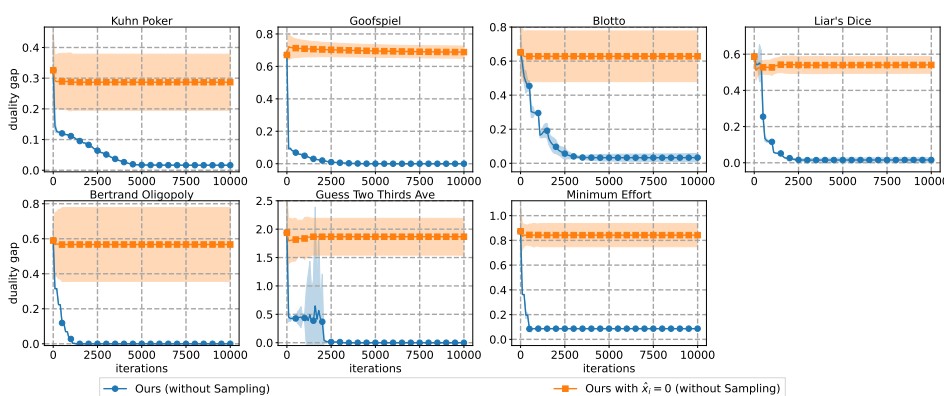

Figure 20: Empirical convergence rates of the algorithms that minimize NAL and NAL with $\hat{\boldsymbol{x}}_i = 0$, respectively, when the optimizer is Adam, the activation function of the final layer is sparsemax (Martins & Astudillo, 2016), and sampling is not used.

Table 3: The hyperparameters of the algorithm that learns an NE via minimizing ADI.

|  | $\eta$ | $\tau$ | $T_u$ | $\alpha$ | $\beta$ |
|---|---|---|---|---|---|
| Kuhn Poker | 0.00001 | 1 | 500 | 0.9 | 0.5 |
| Goofspiel | 0.0001 | 0.1 | 200 | 0.9 | 0.5 |
| Blotto | 0.00001 | 0.1 | 500 | 0.9 | 0.5 |
| Liar's Dice | 0.0001 | 0.1 | 500 | 0.9 | 0.5 |
| Bertrand Oligopoly | 0.00001 | 0.1 | 200 | 0.9 | 0.5 |
| Guess Two Thirds Ave | 0.0001 | 0.1 | 1000 | 0.9 | 0.5 |
| Minimum Effort | 0.00001 | 0.1 | 200 | 0.9 | 0.5 |
| Majority Voting | 0.00001 | 0.1 | 200 | 0.9 | 0.5 |

Table 4: The hyperparameters of the algorithm that learns an NE via minimizing NashApr.

|  | $\eta$ |
|---|---|
| Kuhn Poker | 0.0001 |
| Goofspiel | 0.00001 |
| Blotto | 0.0001 |
| Liar's Dice | 0.0001 |
| Bertrand Oligopoly | 0.00001 |
| Guess Two Thirds Ave | 0.0001 |
| Minimum Effort | 0.0001 |
| Majority Voting | 0.0001 |

