# OpenReview forum: "Learning Nash Equilibria in Normal-Form Games via Approximating Stationary Points"
_ICLR.cc/2025/Conference — Submitted to ICLR 2025_

### Official Review · Reviewer_RHJo · 2024-10-22

**Soundness:** 2
**Presentation:** 3
**Contribution:** 2
**Rating:** 3
**Confidence:** 5

**Summary:**

This paper proposes a novel training signal (function) for solving games that is amenable to unbiased estimation. It has lower variance than prior functions that have been proposed. Experiments show that an update rule derived from this function performs well on normal-form games; in these experiments players’ strategies are represented by overparameterized neural networks and updates can be performed more generally with arbitrary DL optimizers.

**Strengths:**

This paper tackles an interesting problem in computational game theory in trying to develop an improved unbiased estimator of exploitability. Lemma 4.1 in this paper is a key insight and interesting contribution that generalizes the idea of a "projected gradient" from prior work in Gemp et al. 2024. The authors use this to derive a learning algorithm. The proposed algorithm is competitive against baselines in several games from the literature (EFGs converted to NFG form). Also, the idea of sampling actions $a_i'$ from arbitrary distributions (lines 8 & 9 of Algorithm 1) for which to estimate gradients $\nabla_{x,a_i'}$ is novel. Prior work only considered sampling $a_i'$ from uniform.

**Weaknesses:**

I like the goal of this paper, but there are several technical issues: the proposed “loss” is not a loss in the traditional sense of ML; stop-grad operators ruin essential properties of gradients, especially those that suggest stationary points are easy to find; the proposed algorithm lacks novelty— it is similar to basic simultaneous gradient ascent and more closely to magnetic mirror descent; DL has been applied to NE in prior work (GAES [1], NES [2], NFGTransformer [3]).

1) A loss function is generally accepted to be a function whose value captures some measure of sub-optimality and whose global minimum expresses a desired solution. This is not the case with $\mathcal{L}^{\tau}\_{NAL}$. While $\\mathcal{L}^{\\tau}\_{NAL} = 0$ at a Nash equilibrium of the entropy regularized game, $\\mathcal{L}^{\\tau}\_{NAL}$ may be less than 0 for some other strategy profiles. This would suggest there exist profiles that are “closer” to a Nash equilibrium than actual equilibria, which is counter-intuitive. One can confirm this phenomenon with a simple example; consider the game where the row player’s payoff matrix is $\\begin{bmatrix} 0 & 0 \\\\ 1 & 1 \\end{bmatrix}$ and the column player’s is the transpose. If you plot $\\mathcal{L}^{\\tau}\_{NAL}$ for $\\tau=0.2$ and $\\hat{x}_i = \\mathbf{1} / \\vert \\mathcal{A}\_i \\vert = [1/2 ,1/2]$ while varying each player’s mixed strategy $x_r = [p, 1 - p]$ and $x_c = [p, 1 - p]$, you will see the loss is negative for a range of strategies. For this reason, please plot the actual loss function in experiments. I do not doubt the empirical results of Figure 3 that show the difference between the true expected loss and its estimated value; that is expected. But I would not be surprised if a plot of the true expected loss shows negative values at times.

2) Defining loss functions with “stop grad” operators in them generally leads to “pseudo-gradients”, i.e., their gradients no longer satisfy critical properties that gradients normally satisfy. For instance, it is true, as you state, that finding a stationary point of a loss function is easier than finding a global minimum (Jin et al. 2017). However, finding a stationary point of an arbitrary vector field (i.e., a point where a pseudo-gradient is zero) is nontrivial. Note that by searching for a Nash equilibrium of an entropy regularized game, you are computing a quantal response equilibirum (QRE). It is known that computing a QRE is PPAD-hard [4]. And there is another reason you can expect finding a stationary point to be hard which I discuss next.

3) The algorithm proposed is essentially simultaneous gradient ascent but with entropy regularized gradients and decaying temperature coefficients. This is quite similar to prior approaches, e.g., magnetic mirror descent [5]. Also, it has been proven that there exists no continuous update dynamics that converge to Nash equilibria in generic games [6,7], which includes the update proposed in this paper. Lastly, note that if one projects equation (3) onto the tangent space of the simplex, one obtains the projected gradient defined in Gemp et al. 2024: $\\Pi_{T\\Delta}[\\nabla_{x_i} \\mathcal{L}^{\\tau}\_{NAL}] = F_i^{\\tau,x} - \\langle F_i^{\\tau,x}, \\mathbf{1} / \\vert \\mathcal{A}\_i \\vert \\rangle \\mathbf{1} = -\\Pi_{T\\Delta}(\\nabla_x u_i^{\tau})$ regardless of the $\\hat{x}_i$ chosen. Based on this, I would expect the behavior of the update proposed in this paper to behave similarly to simultaneous gradient ascent (with regularization).

4) The proofs of Theorems 4.2 and 4.3 follow from minor modifications of results in Gemp et al. 2024, e.g., Lemmas 2 and 12 therein. Also, how does the duality gap of your approach in Thoerem 4.2 compare to Gemp et al. 2024? Presumably, since $\Pi_{T\Delta}(\\nabla_{x} \mathcal{L}^{\\tau}\_{NAL}) = -\Pi_{T\Delta}(\\nabla_x u_i^{\tau})$, then $||\Pi_{T\Delta}(\\nabla_x u_i^{\tau})|| \le ||\\nabla_{x} \mathcal{L}^{\\tau}\_{NAL}||$, i.e., the bound proposed here is not as tight. Do you agree?

Minor:
- The dot notation is unideal given it’s widely accepted to mean time derivative, $\\dot{x} = dx/dt$. Prior work uses $sg[\\cdot]$ for the "stop grad" operator which is clearer and more explicit. I would suggest writing equation (3) as $\\mathcal{L}^{\\tau}\_{NAL}(\\boldsymbol{x}) = \\sum\_{i \\in \\mathcal{N}} \\langle sg[ F^{\\tau,\boldsymbol{x}}\_i - \\langle F^{\\tau,\\boldsymbol{x}}\_i, \hat{x}_i \\rangle \mathbf{1} ], x_i \\rangle$ with the explicit $sg[\\cdot]$ operator or as $\\mathcal{L}^{\\tau}\_{NAL}(\\boldsymbol{x}) = \\sum\_{i \\in \\mathcal{N}} \\langle F^{\\tau,\\tilde{\\boldsymbol{x}}}\_i - \\langle F^{\\tau,\\tilde{\\boldsymbol{x}}}\_i, \hat{x}_i \\rangle \mathbf{1}, x_i \\rangle$ where the first term in the inner product evaluates $F$ at $\\tilde{\\boldsymbol{x}}$.
- In equation (6), please state the distribution that the expectation is taken w.r.t., i.e., $\\mathbb{E}_{s \\sim p_i(s)}$.
- Theorem 4.2 defines the utility function $u_i^{\\tau}(x) = u_i(x) - \\tau x_i \\log(x_i)$. The second term is currently interpreted as a vector. It is missing a sum or a transpose, i.e., it should be $u_i^{\\tau}(x) = u_i(x) - \\tau x_i^\\top \\log(x_i)$.

[1] Goktas, Denizalp, et al. "Generative Adversarial Equilibrium Solvers." The Twelfth International Conference on Learning Representations.

[2] Marris, Luke, et al. "Turbocharging solution concepts: Solving NEs, CEs and CCEs with neural equilibrium solvers." Advances in Neural Information Processing Systems 35 (2022): 5586-5600.

[3] Liu, Siqi, et al. "NfgTransformer: Equivariant Representation Learning for Normal-form Games." The Twelfth International Conference on Learning Representations.

[4] Milec, David, et al. "Complexity and algorithms for exploiting quantal opponents in large two-player games." Proceedings of the AAAI Conference on Artificial Intelligence. Vol. 35. No. 6. 2021.

[5] Sokota, Samuel, et al. "A Unified Approach to Reinforcement Learning, Quantal Response Equilibria, and Two-Player Zero-Sum Games." The Eleventh International Conference on Learning Representations.

[6] Milionis, Jason, et al. "An impossibility theorem in game dynamics." Proceedings of the National Academy of Sciences 120.41 (2023): e2305349120.

[7] Vlatakis-Gkaragkounis, Emmanouil-Vasileios, et al. "No-regret learning and mixed nash equilibria: They do not mix." Advances in Neural Information Processing Systems 33 (2020): 1380-1391.

**Questions:**

Questions / Suggestions for improvement:
- Lemma 4.1 is interesting. Can you derive the $y \in \\Delta$ that minimizes variance of the norm of $b - \\langle b, y \\rangle \\mathbf{1}$?
- Why, in lines 8 & 9 of Algorithm 1, do you suggest sampling actions from a strategy that interpolates between uniform and the current strategy? Can you provide any analysis that motivates this decision? Note it appears you set $\\epsilon = 1$ in all experiments anyways, and this is precisely the setting that Gemp et al. 2024 already provided analysis for in their Table 2.
- How does your algorithm behave if you set $\\hat{x}_i=0$? How does it behave if you set $\\hat{x}_i = \\mathbf{1} / \\vert \\mathcal{A}_i \\vert$?
- DL is used as a primary motivation for the paper yet it is not discussed until the experiment section. Why does it make sense to overparameterize a strategy (e.g., 2-action game) with a neural network and then run gradient descent on the network parameters?

---

> ### Author Response · Authors · 2024-11-23
>
> We thank the reviewer's comments, this helps us a lot to improve the paper.
>
> **Q1**: DL has been applied to NE in prior work (GAES [1], NES [2], NFGTransformer [3]).
>
> **A**: Using DL is not our contribution. Our innovation lies in proposing a novel loss function for learning the NE via DL with an unbiased estimate.
>
> Additionally, the loss functions in the works [1] and [3] you referenced both include the max operator. According to the analysis in (Gemp et al., 2024), the inclusion of the max operator generally leads to biased estimates under sampled play. Regarding work [2], based on Eq. (9) of [2], the final output strategy involves an exponential operation. Even if we obtain an unbiased estimate of $A_p$ (as defined in [2]), we cannot obtain an unbiased estimate of the strategy itself. In other words, work [2] also encounters biased estimation issues under sampled play. In conclusion, these works you mentioned all suffer from the biased estimation issues under sampled play. In contrary, our work proposes an efficient way to learn the NE with an unbiased estimate.
>
> **Q2**: A loss function is generally accepted to be a function whose value captures some measure of sub-optimality and whose global minimum expresses a desired solution. This is not the case with $L\_{NAL}^τ$. While $L\_{NAL}^τ=0$ at a Nash equilibrium of the entropy regularized game, LNALτ may be less than 0 for some other strategy profiles. This would suggest there exist profiles that are “closer” to a Nash equilibrium than actual equilibria, which is counter-intuitive. One can confirm this phenomenon with a simple example; consider the game where the row player’s payoff matrix is [0011] and the column player’s is the transpose. If you plot $L\_{NAL}^τ$ for $τ=0.2$ and $x_i=1/|A_i|=[1/2,1/2]$ while varying each player’s mixed strategy $x_r=[p,1−p]$ and $x_c=[p,1−p]$, you will see the loss is negative for a range of strategies. For this reason, please plot the actual loss function in experiments. I do not doubt the empirical results of Figure 3 that show the difference between the true expected loss and its estimated value; that is expected. But I would not be surprised if a plot of the true expected loss shows negative values at times.
>
> **A**: You are absolutely correct that this phenomenon that the value of NAL may be negative exists, as a stationary point may indeed be a saddle point. However, this is precisely one of our key ideas: by approximating stationary points instead of pursuing the global minimum, we aim to reduce the difficulty of learning NE while using our loss function.
>
> Regarding your suggestion to "plot the actual loss function in experiments," we have addressed this in our revision (Appendix E, Figure 17, highlighted in red). The value of NAL can indeed be either greater than or less than 0.

---

> > ### Author Response · Authors · 2024-11-23
> >
> > **Q3**: Defining loss functions with “stop grad” operators in them generally leads to “pseudo-gradients”, i.e., their gradients no longer satisfy critical properties that gradients normally satisfy. For instance, it is true, as you state, that finding a stationary point of a loss function is easier than finding a global minimum (Jin et al. 2017). However, finding a stationary point of an arbitrary vector field (i.e., a point where a pseudo-gradient is zero) is nontrivial.
> >
> > **A**: You are correct that using the stop-gradient operator, a widely used technology in DL (Grill et al., 2020;Chen et al., 2021),  indeed introduces "pseudo-gradients."
> >
> > Since the stationary points of NAL are specific to the inclusion of the stop-gradient operator, a feature absent in the definitions of existing loss functions, we acknowledge that it cannot be claimed that learning the stationary points of NAL is easier than finding the global minimum of other loss functions. Therefore, we have removed the previous claim that "this approach improves the tractability by solving stationary points" (highlighted in red). We also revise our statement about tractability, e.g., “inspired by the fact that learning a stationary point is simpler than a global minimum since a global minimum is necessarily a stationary point while a stationary point is not always a global minimum, we ensure that an NE is a stationary point of NAL to improve the computational efficiency“. In other words, we do not claim that learning NE via minimizing NAL is simpler than other loss function in the revisied version.
> >
> > However, the implementation of the existing loss functions in our experiments also necessarily includes the stop-gradient operator. To learn an NE, we only need to locate the stationary points of our loss function, where ${F}^{\tau, {x}}_i - \langle {F}^{\tau, {x}}_i, \hat{{x}}_i \rangle {1} = 0$. In contrast, finding the global minimum of the existing loss functions is required.
> >
> > Specifically, the reason for that the implementation of the existing loss functions includes the stop-gradient operator in Eq. (2) is that $F^{\tau,x}_i$ cannot easily participate in backpropagation. For $F^{\tau,x}_i$ to be involved in backpropagation, it would require exhaustive enumeration of all possible actions of any two players, as described in (Gemp et al., 2024) and (Gemp et al., 2022). This approach is practically infeasible in real-world games, as corroborated by the results in Figures 12 and 13. As shown in the implementation details in Appendix D, all loss functions used in our experiments include the stop-gradient operator.

---

> > > ### Author Response · Authors · 2024-11-23
> > >
> > > **Q4**: Note that by searching for a Nash equilibrium of an entropy regularized game, you are computing a quantal response equilibirum (QRE). It is known that computing a QRE is PPAD-hard [4].
> > >
> > > **A**: We do not aim to solve PPAD-hard problems. To the best of our knowledge, (Gemp et al., 2024) and (Gemp et al., 2022) are also focused on solving QRE. The primary motivation of our paper is to leverage the empirical performance of deep learning (DL) techniques, rather than to propose new theoretical results. At no point do we claim that our contributions offer novel theoretical findings.
> > >
> > > **Q5**: The algorithm proposed is essentially simultaneous gradient ascent but with entropy regularized gradients and decaying temperature coefficients. This is quite similar to prior approaches, e.g., magnetic mirror descent [5].
> > >
> > > **A**: The core of our algorithm lies in identifying the zero point of $F_i^{\tau,x} - \langle F_i^{\tau,x}, \hat{x}_i \rangle 1$, which represents the stationary point of the NAL. This is the key idea of our approach and is fundamentally different from the algorithms you mentioned. The algorithms you referred to use $F_i^{\tau,x}$ as input, but this does not imply that $F_i^{\tau,x}$ being equal at every action corresponds to a stationary point of the neural network. The reason is that we cannot ensure the outputs of the neural network remain invariant, even if the gradients used for backpropagation are identical across neurons in the output layer. We need account for the influence of the neural network.
> > >
> > > Moreover, we address the issue of high variance in existing unbiased loss functions. We propose a fundamentally different approach: instead of ensuring an unbiased estimate of the loss function, we focus on obtaining an unbiased estimate of the gradient.
> > >
> > > **Q6**: Also, it has been proven that there exists no continuous update dynamics that converge to Nash equilibria in generic games [6,7], which includes the update proposed in this paper.
> > >
> > > **A**:
> > >
> > > - Firstly, the conclusions in [6] do not apply to situations involving randomness, as stated in [6]: "*Finally, we note that our impossibility results do not apply to stochastic dynamics—e.g., discrete-time dynamics in which $\varphi(x)$ is a distribution of possible next points*". However, our algorithm involves randomness.
> > >
> > > - Secondly, Yongacoglu et al. (2024) demonstrate that dynamic convergence to a NE exists when $\varphi(x)$ is discontinuous.
> > > - Thirdly, to the best of our knowledge, [7] appears to study the dynamics of FTRL but does not prove the statement "*there exists no continuous update dynamics that converge to Nash equilibria in generic games*".
> > >
> > > **Q7**: Lastly, note that if one projects equation (3) onto the tangent space of the simplex, one obtains the projected gradient defined in Gemp et al. 2024: $\Pi\_{T\Delta}(\nabla\_{x_i}L\_{NAL}^{\tau})=F_i^{τ,x}−⟨F_i^{τ,x},1/|A_i|⟩1=-\Pi\_{T\Delta}(\nabla\_{x_i}u^{\tau}_i)$ regardless of the $x_i$ chosen. Based on this, I would expect the behavior of the update proposed in this paper to behave similarly to simultaneous gradient ascent (with regularization).
> > >
> > > **A**: As we answered in **Q5**: The algorithms you referred to use $F_i^{\tau,x}$ as input, but this does not imply that $F_i^{\tau,x}$ at every action corresponds to a stationary point of the neural network. The reason is that we cannot ensure the outputs of the neural network remain invariant, even if the gradients used for backpropagation are identical across neurons in the output layer. We need account for the influence of the neural network.
> > >
> > > **Q8**:The proofs of Theorems 4.2 and 4.3 follow from minor modifications of results in Gemp et al. 2024, e.g., Lemmas 2 and 12 therein. Also, how does the duality gap of your approach in Thoerem 4.2 compare to Gemp et al. 2024? Presumably, since $\Pi\_{T\Delta}(\nabla\_{x_i}L\_{NAL}^{\tau})=-\Pi\_{T\Delta}(\nabla\_{x_i}u^{\tau}_i)$ , then $||\Pi\_{T\Delta}(\nabla\_{x_i}L\_{NAL}^{\tau})||≤||-\Pi\_{T\Delta}(\nabla\_{x_i}u^{\tau}_i)||$, i.e., the bound proposed here is not as tight. Do you agree?
> > >
> > > **A**: We do not use the proof technique you mentioned. Instead, we employ the tangent residual approach (Cai et al., 2022) as a critical tool in our proof, which is explicitly stated in the line preceding Theorem 4.2. Our proof methodology is entirely distinct from that of (Gemp et al., 2024). However, the resulting bound is identical to theirs. Specifically, our $C_0 = \sqrt{2|\mathcal{N}|}$ corresponds to the $\sqrt{2}$ term in Lemma 2 of (Gemp et al., 2024). Additionally, Theorem 4.3 is derived based on Theorem 4.2.

---

> ### Author Response · Authors · 2024-11-23
>
> **Q9**: The dot notation is unideal given it’s widely accepted to mean time derivative, $\dot{x}=dx/dt$. Prior work uses sg[⋅] for the "stop grad" operator which is clearer and more explicit. I would suggest writing equation (3) as $\mathcal{L}^{\tau}\_{NAL}({x}) = \sum\_{i \in \mathcal{N}} \langle sg[{F}^{\tau, {x}}_i- \langle {F}^{\tau, {x}}_i, \hat{{x}}_i \rangle {1}], {x}_i \rangle,$ with the explicit sg[⋅] operator. In equation (6), please state the distribution that the expectation is taken w.r.t., i.e., $E\_{s \sim p_i(s)}$. Theorem 4.2 defines the utility function ${x}_i \log{{x}_i}$. The second term is currently interpreted as a vector. It is missing a sum or a transpose, i.e., it should be ${x}_i^{\mathrm{T}} \log{{x}_i}$.
>
> **A**: Thank you for your suggestion. We have revised the manuscript accordingly.
>
> **Q10**: Lemma 4.1 is interesting. Can you derive the y∈Δ that minimizes variance of the norm of b−⟨b,y⟩1?
>
> **A**: We apologize if we misunderstood your concern. We assume you are referring to the variance caused by sampling. If this is the case, the minimum variance for any given function is always 0.
>
> **Q11**:Why, in lines 8 & 9 of Algorithm 1, do you suggest sampling actions from a strategy that interpolates between uniform and the current strategy? Can you provide any analysis that motivates this decision? Note it appears you set ϵ=1 in all experiments anyways, and this is precisely the setting that Gemp et al. 2024 already provided analysis for in their Table 2.
>
> **A**: It is the $\epsilon$-greedy exploration strategy, which is commonly used in game theory, reinforcement learning, and online learning. It is designed to balance the trade-off between exploration (testing new actions to discover their potential rewards) and exploitation (selecting actions that are already known to provide high rewards).
>
> The estimation method shown in the last column of Table 2 in Gemp et al. (2024) is the importance sampling method, where the uniform strategy is used as the sampling strategy. The importance sampling method allows any strategy within the interior of the simplex to be used as the sampling strategy. However, it is important to note that in their algorithm implementation, Gemp et al. (2024) adopt the estimation method from Gemp et al. (2022) in order to incorporate ${F}^{\tau, {x}}_i$ into the gradient backpropagation.
>
> **Q12**:How does your algorithm behave if you set $\hat{x}_i=0$? How does it behave if you set $\hat{x}_i=1/|A_i|$?
>
> **A**: Firstly, $\hat{x}_i=0$ is infeasible because Lemma 4.1 requires $\hat{x}_i$ to lie within the simplex. Secondly, in our experiments, we set $\epsilon=1$, which results in $\hat{x}_i=1/|A_i|$ under this condition (line 328-329).
>
> **Q13**: DL is used as a primary motivation for the paper yet it is not discussed until the experiment section.
>
> **A**: Thank you for your suggestion. We have added a section in the Appendix (Appendix A) discussing learning NE via DL (highlighted in red).
>
> Furthermore, in Section 2, we discuss the loss functions that can be utilized for learning NE via DL. In Section 4, we highlight that the key idea of our algorithm is inspired by the observation that "optimizers (Robbins & Monro, 1951; Bottou, 2010; Kingma & Ba, 2014) commonly used in DL require only unbiased estimates of the first-order gradient."  Finally, the problem we aim to address is, in fact, the unbiased estimation issue in DL-based stochastic optimization. All these aspects are closely related to DL.

---

> ### Author Response · Authors · 2024-11-23
>
> **Q14**: Why does it make sense to overparameterize a strategy (e.g., 2-action game) with a neural network and then run gradient descent on the network parameters?
>
> **A**: The scale of the game is determined not only by the number of actions but also by the number of players. For an n-player, m-action, general-sum normal-form game, storing the payoff matrix requires $nm^n$ entries.
>
> As described in the caption of Figure 1 in our original submission, the game settings for Kuhn Poker, Goofspiel, Blotto, Liar's Dice, Bertrand Oligopoly, Guess Two-Thirds Average, Minimum Effort, and Majority Voting are as follows: 2 players with 64 actions, 2 players with 384 actions, 4 players with 66 actions, 2 players with 2304 actions, 4 players with 50 actions, 4 players with 50 actions, 5 players with 30 actions, and 11 players with 5 actions, respectively. The corresponding sizes of the payoff matrices are 8,192 for Kuhn Poker, 294,912 for Goofspiel, 75,898,944 for Blotto, 10,616,832 for Liar's Dice, 25,000,000 for Bertrand Oligopoly, 25,000,000 for Guess Two-Thirds Average, 121,500,000 for Minimum Effort, and 537,109,375 for Majority Voting.
>
> | Game                 | Payoff Matrix Size | Action Size | Players |
> | -------------------- | -----------------: | ----------: | ------: |
> | Kuhn Poker           |              8,192 |          64 |       2 |
> | Goofspiel            |            294,912 |         384 |       2 |
> | Blotto               |         75,898,944 |          66 |       4 |
> | Liar's Dice          |         10,616,832 |       2,304 |       2 |
> | Bertrand Oligopoly   |         25,000,000 |          50 |       4 |
> | Guess Two-Thirds Avg |         25,000,000 |          50 |       4 |
> | Minimum Effort       |        121,500,000 |          30 |       5 |
> | Majority Voting      |        537,109,375 |           5 |      11 |
>
> Notably, Majority Voting has the smallest action size and the largest payoff  matrix size.
>
> The reason for using a deep neural network is its expressive power, allowing it to capture highly complex patterns and relationships that are difficult to model with traditional methods. Neural networks, especially deep architectures, can approximate arbitrary non-linear functions, making them well-suited for representing the strategic decision-making process in game theory. The ability of deep learning models to learn from raw data—without the need for handcrafted features—enables them to uncover intricate equilibrium strategies that might otherwise be overlooked. We use such a neural network demonstrate the power of DL for approximating the NE. Intuitively, a larger neural network is required to better learn the NE in larger games.
>
>
>
> Additional References
>
> [1] Yongacoglu, Bora, et al. "Paths to Equilibrium in Games." *The Thirty-eighth Annual Conference on Neural Information Processing Systems*.
>
> [2] Grill, Jean-Bastien, et al. "Bootstrap your own latent-a new approach to self-supervised learning." *Advances in neural information processing systems* 33 (2020): 21271-21284.
>
> [3] Chen, Xinlei, and Kaiming He. "Exploring simple siamese representation learning." *Proceedings of the IEEE/CVF conference on computer vision and pattern recognition*. 2021.

---

> > ### Comment · Reviewer_RHJo · 2024-11-24
> >
> > Thank you for your response. I have a follow-up question.
> >
> > **Q12**: I understand that you do not suggest setting $\hat{x}_i = 0$ with your Algorithm, but I would still like to see an ablation experiment where you set $\hat{x}_i = 0$. Equivalently, you can overwrite line 15 of Algorithm 1 with $v_i \leftarrow 0$. It would also be good to see these results for both $S=1$ and $S=100$ and compare them to your suggested algorithm.

---

> > > ### Author Response · Authors · 2024-11-26
> > >
> > > The comparison of empirical convergence rates between the standard NAL and  NAL with $\hat{{x}}_i =0 $ is provided in Appendix E (Figure 19). In NAL with $\hat{{x}}_i =0 $, NE is not a stationary point of NAL since the gradient of NAL ${F}^{\tau, {x}}_i - \langle {F}^{\tau, {x}}_i, \hat{{x}}_i \rangle {1}$ is not ${0}$ if ${x}$ is an NE, which may result in the change of parameters of the neural network according to the chain rule, leading to a shift in the strategy. We demonstrate the performance for different values of $S$ ($S=2, 10, 100$). Note that the minimum value of $S$ must be 2 for NAL. If $S=1$, it is not possible to estimate ${F}^{\tau, {x}}_i - \langle {F}^{\tau, {x}}_i, \hat{{x}}_i \rangle {1}$ via Algorithm 1, as the estimated value of ${F}^{\tau, {x}}_i - \langle {F}^{\tau, {x}}_i, \hat{{x}}_i \rangle {1}$ in this case will always be ${0}$ (since $r_i^s - v_i = 0$). From the experimental results, we observe that using NAL with $\hat{{x}}_i = 0$ as the loss function consistently performs worse than using the standard NAL, especially as $S$ decreases. This further validates the effectiveness of setting NE as the stationary point of NAL.

---

> > > > ### Comment · Reviewer_RHJo · 2024-11-26
> > > >
> > > > If $S=1$, then, by line 15, $v_i = r_i / S$ and the estimated value of $g_i$ on line 18 will not be zero. Please check the math.
> > > >
> > > > The experimental results generally confirm my suspicions. For increasing $S$, it appears the algorithms are performing more and more similarly. And for small $S$, your algorithm shows much lower variance. These experiments support my explanation that the primary difference between your method with and without $\hat{x}_i$ is variance of the estimator, and not the actual update direction.

---

> > > > > ### Author Response · Authors · 2024-11-30
> > > > >
> > > > > Thank you for your valuable feedback. Below are our responses to your concerns.
> > > > >
> > > > > **Q1**: If $S=1$, then, by line 15, $v_i = r_i / S$ and the estimated value of $g_i$ on line 18 will not be zero. Please check the math.
> > > > >
> > > > > **A**: As you pointed out, when $S=1$, line 15 indicates that $v_i = \frac{r_i}{S} = r_i$. In addition, the value of $r^s_i$ on line 18 is simply $r_i$ on line 15 because $S=1$, and therefore $r^s_i - v_i = 0$, which implies that $g^s_i = 0$.
> > > > >
> > > > >
> > > > >
> > > > > **Q2**: The experimental results generally confirm my suspicions. For increasing $S$, it appears the algorithms are performing more and more similarly. And for small $S$, your algorithm shows much lower variance. These experiments support my explanation that the primary difference between your method with and without $\hat{x}_i$ is variance of the estimator, and not the actual update direction.
> > > > >
> > > > > **A**: We identify the cause of the behavior you mention, "For increasing $S$, it appears the algorithms are performing more and more similarly." This behavior arises from the softmax function in the final layer of our neural network. Specifically, it is caused by the normalization operation in softmax (i.e., for any $z > 0$, it outputs $z/\text{sum}(z)$, which lies in the simplex), leading to identical parameter updates under ${F}^{\tau, {x}}_i - \langle {F}^{\tau, {x}}_i, \hat{{x}}_i \rangle {1}$ and ${F}^{\tau, {x}}_i$ (without sampling).
> > > > >
> > > > > To address this issue, we replace the softmax function with sparsemax (Martins & Astudillo, 2016), which ensures that the final output remains within the simplex without requiring the normalization step. We conduct additional experiments in which we substitute softmax with sparsemax and vary the parameter $S$ across values $S=2, 10, 100, \infty$, where $S=\infty$ indicates that we compute the true values of ${F}^{\tau, {x}}_i$ and ${F}^{\tau, {x}}_i - \langle {F}^{\tau, {x}}_i, \hat{{x}}_i \rangle {1}$ directly, instead of estimating them. The results can be found in https://www.mediafire.com/file/wn714czculjdiki/ICLR_2025_rebuttal.pdf/file (the results for $S=\infty$ without Majority Voting are also available in Figure 20 of the latest version of our paper). We observe that the standard NAL outperforms the NAL with $\hat{{x}}_i = 0$ by a significant margin. In fact, we never observe convergence for the NAL with $\hat{{x}}_i = 0$. This suggests that when the activation function is softmax, the convergence of NAL with $\hat{{x}}_i = 0$ is likely caused by the identical parameter updates between ${F}^{\tau, {x}}_i - \langle {F}^{\tau, {x}}_i, \hat{{x}}_i \rangle {1}$ and ${F}^{\tau, {x}}_i$ (without sampling). Note that when $S = \infty$, the variance is $0$ because we use the true values of $F^{\tau, {x}}_i$ and $F^{\tau, {x}}_i - \langle F^{\tau, {x}}_i, \hat{x}_i \rangle 1$. In this case, the standard NAL still significantly outperforms the NAL with $\hat{x}_i = 0$ (we do not observe convergence for the NAL with $\hat{{x}}_i = 0$). Therefore, the superior performance of the standard NAL over the NAL with $\hat{x}_i = 0$ is due to "the actual update" rather than "the variance of the estimator."
> > > > >
> > > > >
> > > > >
> > > > > Additional References:
> > > > >
> > > > > Andre Martins and Ramon Astudillo. From softmax to sparsemax: A sparse model of attention and multi-label classification. In Proceedings of the 33rd International conference on machine learning, pp. 1614–1623. PMLR, 2016.

---

> > > > > > ### Comment · Reviewer_RHJo · 2024-12-02
> > > > > >
> > > > > > Q1. Of course. My mistake. Thank you for checking.
> > > > > >
> > > > > > Q2. Yes, that makes sense. The softmax is invariant to constant offsets, i.e., softmax($z$) = softmax($z + c \\mathbf{1}$) with $c \\in \\mathbb{R}$. The results with the sparsemax are interesting, but I think require a complete review in a resubmission if that's the direction you want to move the paper towards.

---

> > > > > > > ### Author Response · Authors · 2024-12-02
> > > > > > >
> > > > > > > We use sparsemax merely to demonstrate that the primary difference between our method with and without $\hat{x}_i$ is  the actual update direction, not the variance of the estimator. This does not alter the main focus of our paper. The specific choice of network is not the focus of our work. The main contribution of our paper lies in introducing a new loss function enabling unbiased sampling, which is applicable to any network architecture. If you have any further questions, we are more than happy to engage in further discussion.

---

> > > > > > > > ### Comment · Reviewer_RHJo · 2024-12-02
> > > > > > > >
> > > > > > > > I understand that $F\_i^{\\tau,x}$ and $F\_i^{\\tau,x} - \\langle F\_i^{\\tau,x}, \\hat{x}_i \\rangle \mathbf{1}$ are different update directions if that's what you mean. However, as I said before, these directions are equal after projecting them onto the tangent space of the simplex, i.e., both become $F\_i^{\\tau,x} - \\frac{1}{m} \\langle F\_i^{\\tau,x}, \\mathbf{1} \\rangle \mathbf{1}$. In some sense, from the perspective of the simplex, they are the same direction. As you have found, the softmax preserves their equivalence as well (which also appears when performing "entropic mirror ascent"). Another way to see they are equal (with respect to updates on the simplex) is to compute (euclidean) projected gradient steps with the two different directions and find they produce the same result. Both entropic mirror ascent and euclidean projected gradient ascent are well understood methods from convex optimization. In contrast, the sparsemax is not and its effect on transforming any two update directions is not well understood. In short, I see that the sparsemax generates different results for the two different directions, but that doesn't further my understanding much beyond the fact that they are different directions and applying an exotic nonlinear operator to them would give different results.

---

> > > > > > > > > ### Author Response · Authors · 2024-12-02
> > > > > > > > >
> > > > > > > > > You mentioned "entropic mirror ascent" and "euclidean projected gradient," both of which are tabular algorithms. However, we focus on scenarios involving neural networks. As demonstrated in our experiments, NAL performs well across different activation functions, while the version of NAL without $\langle {F}^{\tau, {x}}_i, \hat{{x}}_i \rangle {1}$ (you mentioned) performs poorly in networks with a sparsemax activation function. Through our loss function, we are able to design novel network architectures that can more efficiently learn NE. These new networks may differ significantly from existing architectures and do not necessarily rely on the softmax function as the final activation. In addition, these networks even do not backpropagate gradients starting from the final output layer corresponding to $x_i$. Instead, they will backpropagate from a shallower layer. Therefore, the statement about "projecting them onto the tangent space of the simplex" is not valid, as the output of this layer may not lie within the simplex at all.
> > > > > > > > >
> > > > > > > > > Additionally, it is important to note that the network architecture is not central to the contributions of this paper, nor is it the primary focus. Specifically, the impact of different network structures (such as the choice of activation functions) on the update of network parameters is not the focus of this work. Our contribution lies in proposing a novel lower variance unbiased loss function called NAL. We innovate by using the unbiased estimate of the gradient, rather than the value of the loss function itself, to simultaneously achieve both lower variance and an unbiased estimate. Figures 2, 7, and 8 demonstrate that the variance of NAL is significantly lower than that of the only known unbiased loss functions. Figures 1, 4, 5, 11, 13, 14, 15, 16, 19, and 20 show that the convergence rate of learning NE through finding the stationary point of NAL is much faster than that of existing loss functions. Figure 20 further illustrates that our loss function can adapt to different network architectures.

---

### Official Review · Reviewer_PN7y · 2024-11-03

**Soundness:** 2
**Presentation:** 3
**Contribution:** 2
**Rating:** 3
**Confidence:** 3

**Summary:**

The paper investigates the problem of learning Nash equilibrium in large-scale normal-form games using gradient methods. Given that the game matrix can be enormous, estimating the loss function from samples is essential. By leveraging the fact that, at an interior Nash equilibrium, the deviations of all pure actions of a player should be equal, the authors introduce an additional entropy term into the game utility to ensure the modified game has an interior NE. They then construct a loss function whose first-order gradient can be unbiasedly estimated through importance sampling. Theoretical results demonstrate that the approximate NE of the modified game is also an approximate solution for the original game. Experimental results further showcase the effectiveness and low variance of this loss function.

**Strengths:**

1. The proposed approach is well-motivated, as it is often difficult to obtain the entire utility matrix, making loss estimation from sampling necessary.
2. Compared to previous relevant work (Gemp et al., 2024), the proposed loss function significantly reduces variance.
3. The authors conduct extensive experiments to demonstrate the efficacy of their approach, showing that the proposed loss function achieves better approximation and lower variance.

**Weaknesses:**

1. While the experimental results are promising and show a significant reduction in variance compared to Gemp et al. (2024), I find the optimization process of the approach to be unusual and lacking theoretical insights. The proposed loss function $\mathcal{L}_{NAL}^\tau(\mathbf{x}) $ is provided in Equation (2),
but its gradient does not match the expression $ \langle F_i^{\tau, x} - \langle F_i^{\tau, x},\hat{x}_i \rangle \mathbf{1}, x_i\rangle $ in Equation (3).
The authors derive the gradient by stopping the backpropagation of gradients in $F_i^{\tau, x}$.
While such an operation can be easily implemented, it lacks mathematical justification.
The expression in Equation (3) is not necessarily the direction that maximizes the rate of increase for the proposed loss.
Therefore, I do not believe that optimizing the loss using this incorrect gradient is theoretically sound.
2. When discussing Gemp et al. (2024), the authors state that "learning an NE through this loss function requires finding a global minimum in a non-convex optimization problem, which is widely acknowledged as significantly challenging." However, I did not find that the paper addresses this challenge. We still need to locate a global minimum for $ \mathcal{L}_{NAL}^\tau(\mathbf{x}) $.

**Questions:**

What is the input to the neural network in your experiments? Is it a constant 1024-dimensional vector? If so, why utilize a neural network? It seems that using a neural network may be redundant in this case.

---

> ### Author Response · Authors · 2024-11-23
>
> Thank you very much for your review. Here are our replies to your concern.
>
> **Q1**: While the experimental results are promising and show a significant reduction in variance compared to Gemp et al. (2024), I find the optimization process of the approach to be unusual and lacking theoretical insights. The proposed loss function $L\_{NAL}^τ(x)$ is provided in Equation (2), but its gradient does not match the expression $F_i^{τ,x}−⟨F_i^{τ,x},\hat{x}_i⟩1$ in Equation (3). The authors derive the gradient by stopping the backpropagation of gradients in $F^{\tau,x}_i$. While such an operation can be easily implemented, it lacks mathematical justification. The expression in Equation (3) is not necessarily the direction that maximizes the rate of increase for the proposed loss. Therefore, I do not believe that optimizing the loss using this incorrect gradient is theoretically sound. When discussing Gemp et al. (2024), the authors state that "learning an NE through this loss function requires finding a global minimum in a non-convex optimization problem, which is widely acknowledged as significantly challenging." However, I did not find that the paper addresses this challenge. We still need to locate a global minimum for $L\_{NAL}^τ(x)$.
>
> **A**:
>
> - We are indeed searching for the stationary point of the NAL. In the definition of NAL (Eq. (2)), we utilize the stop-gradient operator, a widely used technology in DL (Grill et al., 2020;Chen et al., 2021),  which implies that the term inside this operator does not participate in gradient backpropagation. Following Reviewer RHJo's suggestion, we have updated Equation (2) to $\mathcal{L}^{\tau}\_{NAL}({x}) = \sum\_{i \in \mathcal{N}} \langle sg[{F}^{\tau, {x}}_i - \langle {F}^{\tau, {x}}_i, \hat{{x}}_i \rangle {1}], {x}_i \rangle$ for better clarity (the changes are highlighted in red), where $sg[\cdot]$ denotes the stop-gradient operator. Since $sg[{F}^{\tau, {x}}_i - \langle {F}^{\tau, {x}}_i, \hat{{x}}_i \rangle {1}]$ is excluded from gradient backpropagation, it can be treated as a constant. Consequently, the gradient of NAL is as presented in Eq. (3).
>
> - By applying the stop-gradient operator, we effectively transform the problem of estimating the value of the loss function into the task of estimating its gradient, thereby mitigating the high variance issue observed in (Gemp et al., 2024). When Eq. (3) equals zero, the gradient of NAL is also zero, indicating that the point is a stationary point of NAL. Therefore, we are leanring a stationar point of NAL rather than a global minimum.
>
> - The stationary points of NAL are specific to the inclusion of the stop-gradient operator, a feature absent in the definitions of existing loss functions. In this regard, we acknowledge that it cannot be claimed that learning the stationary points of NAL is easier than finding the global minimum of other loss functions. Therefore, we have removed the previous claims that "this approach improves the tractability by solving stationary points" (highlighted in red) and addressing the challenging "learning an NE through this loss function requires finding a global minimum in a non-convex optimization" you mentioned. We also revise our statement about tractability, e.g., “inspired by the fact that learning a stationary point is simpler than a global minimum since a global minimum is necessarily a stationary point while a stationary point is not always a global minimum, we ensure that an NE is a stationary point of NAL to improve the computational efficiency“. In other words, we do not claim that learning NE via minimizing NAL is simpler than other loss function in the revisied version.
>
> - However, it is worth emphasizing that while other loss functions do not include the stop-gradient operator in their definitions, in practice, these loss functions must employ the stop-gradient operator when solving real-world games. This is because $F^{\tau,x}_i$ cannot feasibly participate in gradient backpropagation. Enabling $F^{\tau,x}_i$ to participate in backpropagation would require iterating over all action pairs for every two players, as done in (Gemp et al., 2024) and (Gemp et al., 2022), which is practically infeasible in real-world games. Our results in Figure 12 and Figure 13 further validate this (Note that the algorithm "Gemp et al., 2024 with the sampling method of Gemp et al." in Figure 13 is identical to the original algorithm design presented in (Gemp et al., 2024)). In fact, as detailed in Appendix D, all the loss functions used in all our experiments (Section 5 Experiments) incorporate the stop-gradient operator.

---

> > ### Author Response · Authors · 2024-11-23
> >
> > **Q2**: What is the input to the neural network in your experiments? Is it a constant 1024-dimensional vector? If so, why utilize a neural network? It seems that using a neural network may be redundant in this case.
> >
> > **A**:
> >
> > - Yes, the input to the neural network is as you described. However, any input is feasible, and this setup is used in our paper primarily for demonstration purposes. For example, in solving real-world games, the input could be a natural language description of the game. Specifically, the natural language description can be processed by a Large Language Model (LLM), and the output from the embedding layer of the LLM can then be used as the input to the network that solves for the NE.
> >
> > - The reason for using a deep neural network is its expressive power, allowing it to capture highly complex patterns and relationships that are difficult to model with traditional methods. Neural networks, especially deep architectures, can approximate arbitrary non-linear functions, making them well-suited for representing the strategic decision-making process in game theory. The ability of deep learning models to learn from raw data—without the need for handcrafted features—enables them to uncover intricate equilibrium strategies that might otherwise be overlooked. We use such a neural network to demonstrate the power of DL for approximating the NE.
> >
> >
> >
> > Additional References:
> >
> > [1] Grill, Jean-Bastien, et al. "Bootstrap your own latent-a new approach to self-supervised learning." *Advances in neural information processing systems* 33 (2020): 21271-21284.
> >
> > [2] Chen, Xinlei, and Kaiming He. "Exploring simple siamese representation learning." *Proceedings of the IEEE/CVF conference on computer vision and pattern recognition*. 2021.

---

> > > ### Author Response · Authors · 2024-12-01
> > >
> > > Dear Reviewer PN7y,
> > >
> > > We sincerely appreciate the time and effort you have devoted to reviewing our paper “6366: Learning Nash Equilibria in Normal-Form Games via Approximating Stationary Points”. We are writing to follow up on the status of your review for our paper, as the Author-Reviewer Discussion phase is coming to a close in less than a week. We hope that if you are satisfied with our answers, you could consider adjusting your score accordingly.
> > >
> > > Best regards,
> > >
> > > The Authors

---

### Official Review · Reviewer_TkxX · 2024-11-04

**Soundness:** 3
**Presentation:** 3
**Contribution:** 3
**Rating:** 6
**Confidence:** 4

**Summary:**

This paper solves the problem of approximating Nash Equilibrium in multi-players normal form games (NFGs). The authors adopts a neural network architecture, that outputs the strategy profile of the players. The goal is to train the network, such that the network outputs are close to the solution concept of Nash Equilibrium, measured by Duality Gap.

The authors propose Nash Advantage Loss (NAL) to train the neural network. The advantage of NAL lies in two fold: 1) NAL has an unbiased estimator, while the only loss function that has this property in the literature is the loss function proposed by (Gemp et al., 2024); 2) compared with (Gemp et al., 2024), the unbiased estimator of NAL achieves lower variance.

**Strengths:**

**Technical Strengths:**

1. The proposed loss function and its corresponding training algorithm (mainly about extracting the unbiased estimators) are novel. This approach also has the potential to approximate the NE of large-scale games.
2. The analysis of the relation between NAL and the NE measure (Duality Gap) as well as the analysis of the variance comparison between NAL and (Gemp et al., 2024) are solid, providing theoretical guarantee of NAL.

**Other Strengths:**

1. The presentation of the paper is logically coherent and the derivation is also detailed, making the paper easy to follow.
2. The authors conduct sufficient experiments and analysis, showing the applicability of the results.

**Weaknesses:**

1. It seems unfair to compare the variance of NAL with the variance of loss in (Gemp et al., 2024). It is because the gradient of NAL seems to have square root order magnitudes compared with the gradient of loss in (Gemp et al., 2024).
2. It also seems inappropriate to say this method solves the stationary point of NAL. It is essentially defining a "gradient" in equation (3) for back propagation, which is not the full gradient of the NAL function. The goal is equivalent to finding the zero point of equation(3). Therefore, the claim that this approach improves the tractability by solving stationary point instead of global minimum is questionable.
3. The Lemma 4.1 is in a slightly abrupt position.
4. In Figure 5, the duality gap of NashApr is too strange. Maybe the learning rate is not large enough when the optimizer is SGD.

**Questions:**

1. Does there exists a well-defined function whose gradient is exactly equation (3)?
2. I'm confused about whether the neural network takes game representation as inputs in the experiments. In Line 45, the authors mention that the game inputs is exponentially large, so direct computing methods (without estimate) suffer from large computational costs. Why do you train a neural network to output the strategy profile rather than optimize on strategy profile $x$ directly?
3. What is the problem size of the NFGs in the paper's experiments?

---

> ### Author Response · Authors · 2024-11-23
>
> We hope that these comments have addressed the reviewer’s concern.
>
> **Q1**: It seems unfair to compare the variance of NAL with the variance of loss in (Gemp et al., 2024). It is because the gradient of NAL seems to have square root order magnitudes compared with the gradient of loss in (Gemp et al., 2024).
>
> **A**: 1) For the calculation of the variance of our loss function values and the loss function values in (Gemp et al., 2024), both the estimated and true values need to be computed. Unfortunately, the loss function in (Gemp et al., 2024) must include a squared term, otherwise, their loss cannot be estimated unbiasedly. Moreover, it is infeasible to compute the variance by taking the square root of the estimated value of loss in (Gemp et al., 2024) after the estimated value was computed because the estimated value may be negative, making the square root operation impossible. 2) Furthermore,  directly taking the square root of the variance of the loss in (Gemp et al., 2024) is incorrect. This operation does not yield the variance of the loss in (Gemp et al., 2024) without the square. (We present a comparison of the variance of NAL with the square root of the variance of the loss function from Gemp et al. (2024) in Figure 18. We observe that the variance of NAL is, in most cases, significantly smaller than the square root of the variance of the loss function presented in Gemp et al. (2024). )
>
>
>
> **Q2**: It also seems inappropriate to say this method solves the stationary point of NAL. It is essentially defining a "gradient" in equation (3) for back propagation, which is not the full gradient of the NAL function. The goal is equivalent to finding the zero point of equation(3). Therefore, the claim that this approach improves the tractability by solving stationary point instead of global minimum is questionable.
>
> **A**:
>
> - We are indeed searching for the stationary point of the NAL. In the definition of NAL (Eq. (2)), we utilize the stop-gradient operator, a widely used technology in DL (Grill et al., 2020;Chen et al., 2021), which implies that the term inside this operator does not participate in gradient backpropagation. Following Reviewer RHJo's suggestion, we have updated Equation (2) to $\mathcal{L}^{\tau}\_{NAL}({x}) = \sum\_{i \in \mathcal{N}} \langle sg[{F}^{\tau, {x}}_i - \langle {F}^{\tau, {x}}_i, \hat{{x}}_i \rangle {1}], {x}_i \rangle$ for better clarity (the changes are highlighted in red), where $sg[\cdot]$ denotes the stop-gradient operator. Since $sg[{F}^{\tau, {x}}_i - \langle {F}^{\tau, {x}}_i, \hat{{x}}_i \rangle {1}]$ is excluded from gradient backpropagation, it can be treated as a constant. Consequently, the gradient of NAL is as presented in Eq. (3). By applying the stop-gradient operator, we effectively transform the problem of estimating the value of the loss function into the task of estimating its gradient, thereby mitigating the high variance issue observed in (Gemp et al., 2024). When Eq. (3) equals zero, the gradient of NAL is also zero, indicating that the point is a stationary point of NAL.
> - The stationary points of NAL are specific to the inclusion of the stop-gradient operator, a feature absent in the definitions of existing loss functions. In this regard, we acknowledge that it cannot be claimed that learning the stationary points of NAL is easier than finding the global minimum of other loss functions. Therefore, we have removed the previous claim that "this approach improves the tractability by solving stationary points" (highlighted in red). We also revise our statement about tractability, e.g., “inspired by the fact that learning a stationary point is simpler than a global minimum since a global minimum is necessarily a stationary point while a stationary point is not always a global minimum, we ensure that an NE is a stationary point of NAL to improve the computational efficiency“. In other words, we do not claim that learning NE via minimizing NAL is simpler than other loss function in the revisied version.
> - However, it is worth emphasizing that while other loss functions do not include the stop-gradient operator in their definitions, in practice, these loss functions must employ the stop-gradient operator when solving real-world games. This is because $F^{\tau,x}_i$ cannot feasibly participate in gradient backpropagation. Enabling $F^{\tau,x}_i$ to participate in backpropagation would require iterating over all action pairs for every two players, as done in (Gemp et al., 2024) and (Gemp et al., 2022), which is practically infeasible in real-world games. Our results in Figure 12 and Figure 13 further validate this (Note that the algorithm "Gemp et al., 2024 with the sampling method of Gemp et al." in Figure 13 is identical to the original algorithm design presented in (Gemp et al., 2024)). In fact, as detailed in Appendix D, all the loss functions used in our all experiments incorporate the stop-gradient operator.

---

> > ### Author Response · Authors · 2024-11-23
> >
> > **Q3**: In Figure 5, the duality gap of NashApr is too strange. Maybe the learning rate is not large enough when the optimizer is SGD.
> >
> > **A**: In the revision, we include results using NashApr as the loss function with the optimizer set to SGD and the learning rate increased by 10x and 100x compared to the learning rates shown in Appendix F (highlighted in red in Appendix E). Specifically, the learning rates in Appendix F are range from 1e-5 to 1e-4, while our new results use the learning rates from 1e-4 to 1e-2. Notably, even with these increased learning rates, NashApr continues to perform very poorly.
> >
> > **Q4**: Does there exists a well-defined function whose gradient is exactly equation (3)?
> >
> > **A**: We apologize that without employing the stop-gradient operator, this function may not exist.
> >
> > **Q5**: I'm confused about whether the neural network takes game representation as inputs in the experiments. In Line 45, the authors mention that the game inputs is exponentially large, so direct computing methods (without estimate) suffer from large computational costs. Why do you train a neural network to output the strategy profile rather than optimize on strategy profile x directly?
> >
> > **A**:
> >
> > In the experiment, the neural network's input is a constant 1024-dimensional vector rather than a representation of the game itself. This approach is essentially equivalent to the "optimize on strategy profile x directly" method you referred to. Specifically, "optimize on strategy profile x directly" can be interpreted as training a neural network with a single layer to output the strategy profile. Intuitively, a multi-layer neural network has greater expressive power and can potentially learn the NE more efficiently.
> >
> > However, any input is feasible, and this setup is used in our paper primarily for demonstration purposes. For example, in solving real-world games, the input could be a natural language description of the game. Specifically, the natural language description can be processed by a Large Language Model (LLM), and the output from the embedding layer of the LLM can then be used as the input to the network that solves for the NE.
> >
> > Additionally, since the payoff matrix grows exponentially with the size of the game, sampling is still required to update the strategy profile. Without sampling, the computational complexity would inevitably grow exponentially for the update.
> >
> > **Q6**: What is the problem size of the NFGs in the paper's experiments?
> >
> > **A**: As described in the caption of Figure 1 in our original submission, the game settings for Kuhn Poker, Goofspiel, Blotto, Liar's Dice, Bertrand Oligopoly, Guess Two-Thirds Average, Minimum Effort, and Majority Voting are as follows: 2 players with 64 actions, 2 players with 384 actions, 4 players with 66 actions, 2 players with 2304 actions, 4 players with 50 actions, 4 players with 50 actions, 5 players with 30 actions, and 11 players with 5 actions, respectively. The corresponding sizes of the payoff matrices are 8,192 for Kuhn Poker, 294,912 for Goofspiel, 75,898,944 for Blotto, 10,616,832 for Liar's Dice, 25,000,000 for Bertrand Oligopoly, 25,000,000 for Guess Two-Thirds Average, 121,500,000 for Minimum Effort, and 537,109,375 for Majority Voting.
> >
> >
> > | Game                 | Payoff Matrix Size | Action Size | Players |
> > | -------------------- | -----------------: | ----------: | ------: |
> > | Kuhn Poker           |              8,192 |          64 |       2 |
> > | Goofspiel            |            294,912 |         384 |       2 |
> > | Blotto               |         75,898,944 |          66 |       4 |
> > | Liar's Dice          |         10,616,832 |       2,304 |       2 |
> > | Bertrand Oligopoly   |         25,000,000 |          50 |       4 |
> > | Guess Two-Thirds Avg |         25,000,000 |          50 |       4 |
> > | Minimum Effort       |        121,500,000 |          30 |       5 |
> > | Majority Voting      |        537,109,375 |           5 |      11 |
> >
> >
> >
> > Additional References:
> >
> > [1] Grill, Jean-Bastien, et al. "Bootstrap your own latent-a new approach to self-supervised learning." *Advances in neural information processing systems* 33 (2020): 21271-21284.
> >
> > [2] Chen, Xinlei, and Kaiming He. "Exploring simple siamese representation learning." *Proceedings of the IEEE/CVF conference on computer vision and pattern recognition*. 2021.

---

> > > ### Comment · Reviewer_TkxX · 2024-12-02
> > >
> > > Thanks for your response, but I will probably keep my evaluation. Although I appreciate the experiments, I still do not accept the claim about  "stationary point". NAL looks more like an algorithm instead of a loss function, which outputs a decending direction.

---

> > > > ### Author Response · Authors · 2024-12-03
> > > >
> > > > Thank you very much for your response. We agree with you that NAL is not a common loss function due to the stop gradient. However, we would like to clarify that, according to the definition of a stationary point, "In mathematics, particularly in calculus, a stationary point of a differentiable function of one variable is a point on the graph of the function where the function's derivative is zero"[1], the NE is indeed the stationary point of the NAL.
> > > >
> > > > [1] https://en.wikipedia.org/wiki/Stationary_point

---

### Official Review · Reviewer_BQo9 · 2024-11-20

**Soundness:** 2
**Presentation:** 3
**Contribution:** 2
**Rating:** 3
**Confidence:** 4

**Summary:**

The paper studies the problem of computing approximate Nash equilibria. In particularly large games, it is important to be able to run algorithms based on gradient samples similar to standard techniques applied for training NNs. This paper builds upon the recent work of Gemp et al that produced a specific loss function tailored for this task that allowed for unbiased stochastic optimization. This paper defines an alternative loss function that they call Nash Advantage Loss (NAL), which they use for learning Nash equilibria in normal-form games using deep learning. NAL is also unbiased and exhibits lower variance than the pre-existing methodology by Gemp et al.  Furthermore, the author claim that learning a NE by minimizing NAL is more tractable, as an NE is a stationary point of NAL rather than having to be a global minimum as it was required by the previous technique. The paper inncludes experimental results that show the effectiveness of the approach.

**Strengths:**

- The paper studies an exciting problem of solving for general games using combination of ideas from non-convex optimization, regularization and NNs.
- Compared to the previous methodology of Gemp et al the proposed technique shows reduced variance.
- The technique is validated with experimental results.

**Weaknesses:**

- The use of stop-gradients in going from the loss function defined in Equation 2 to Equation 3 does not feel perfectly theoretically justified. For example, it is not clear whether there exists a function whose gradient is exactly Equation 3? But if that is not the case, then it is not clear whether the proposed technique is akin to finding local optima of non-convex landscape or more like the more complex problem of finding zeros of more arbitrary vector field. For the second case, there exist impossibility results [1] that show that local learning rules although clearly sufficient for solving the first problem do not suffice to solve the second one. In any case, I believe that these connections need to be ironed out and expanded more formally.
- The input, parametrization and usage of NNs should be expanded further.
- The paper lacks sufficient comparison/discussion to related prior work. E.g. [2-4]

[1] Milionis, et al. "An impossibility theorem in game dynamics." PNAS 2024
[2] Marris et al. "Turbocharging solution concepts: Solving NEs, CEs and CCEs with neural equilibrium solvers." NeurIPS 2
[3]  Goktas et al. "Generative Adversarial Equilibrium Solvers." ICLR 24
[4] Liu et al. "NfgTransformer: Equivariant Representation Learning for Normal-form Games." ICLR 24

**Questions:**

Is Equation 3 the gradient of some precise loss function?
If not, why is it appropriate to consider your approach as finding stationary points?

---

> ### Author Response · Authors · 2024-11-23
>
> We thank the reviewer for the insightful and useful feedback, please see the following for our response.
>
> **Q1**: The use of stop-gradients in going from the loss function defined in Equation 2 to Equation 3 does not feel perfectly theoretically justified. For example, it is not clear whether there exists a function whose gradient is exactly Equation 3? But if that is not the case, then it is not clear whether the proposed technique is akin to finding local optima of non-convex landscape or more like the more complex problem of finding zeros of more arbitrary vector field. For the second case, there exist impossibility results [1] that show that local learning rules although clearly sufficient for solving the first problem do not suffice to solve the second one. In any case, I believe that these connections need to be ironed out and expanded more formally.
>
> **A**:
>
> - Firstly, [1] does not investigate the "problem of finding zeros of more arbitrary vector fields." Instead, [1] studies whether there exist continuous update dynamics that converge to the NE. However, the conclusions in [1] do not apply to situations involving randomness, as stated in [1]: "*Finally, we note that our impossibility results do not apply to stochastic dynamics—e.g., discrete-time dynamics in which $\varphi(x)$ is a distribution of possible next points*". Our algorithm involves randomness. In addition, Yongacoglu et al. (2024) demonstrate that dynamic convergence to a NE exists when $\varphi(x)$ is discontinuous.
>
> - Secondly, we apologize for the fact that without employing the stop-gradient operator, the function may not exist. However, we are indeed searching for the stationary point of the NAL. In the definition of NAL (Eq. (2)), we utilize the stop-gradient operator, a widely used technology in DL (Grill et al., 2020;Chen et al., 2021), which implies that the term inside this operator does not participate in gradient backpropagation. Following Reviewer RHJo's suggestion, we have updated Equation (2) to $\mathcal{L}^{\tau}\_{NAL}({x}) = \sum\_{i \in \mathcal{N}} \langle sg[{F}^{\tau, {x}}_i - \langle {F}^{\tau, {x}}_i, \hat{{x}}_i \rangle {1}], {x}_i \rangle$ for better clarity (the changes are highlighted in red), where $sg[\cdot]$ denotes the stop-gradient operator. Since $sg[{F}^{\tau, {x}}_i - \langle {F}^{\tau, {x}}_i, \hat{{x}}_i \rangle {1}]$ is excluded from gradient backpropagation, it can be treated as a constant. Consequently, the gradient of NAL is as presented in Eq. (3). By applying the stop-gradient operator, we effectively transform the problem of estimating the value of the loss function into the task of estimating its gradient, thereby mitigating the high variance issue observed in (Gemp et al., 2024). When Eq. (3) equals zero, the gradient of NAL is also zero, indicating that the point is a stationary point of NAL.
> - Thirdly, since the stationary points of NAL are specific to the inclusion of the stop-gradient operator, a feature absent in the definitions of existing loss functions, we acknowledge that it cannot be claimed that learning the stationary points of NAL is easier than finding the global minimum of other loss functions. Therefore, we have removed the previous claim that "this approach improves the tractability by solving stationary points" (highlighted in red). We also revise our statement about tractability, e.g., “inspired by the fact that learning a stationary point is simpler than a global minimum since a global minimum is necessarily a stationary point while a stationary point is not always a global minimum, we ensure that an NE is a stationary point of NAL to improve the computational efficiency“. In other words, we do not claim that learning NE via minimizing NAL is simpler than other loss function in the revisied version.
> - Lastly, it is worth emphasizing that while other loss functions do not include the stop-gradient operator in their definitions, in practice, these loss functions must employ the stop-gradient operator when solving real-world games. This is because $F^{\tau,x}_i$ cannot feasibly participate in gradient backpropagation. Enabling $F^{\tau,x}_i$ to participate in backpropagation would require iterating over all action pairs for every two players, as done in (Gemp et al., 2024) and (Gemp et al., 2022), which is practically infeasible in real-world games. Our results in Figure 12 and Figure 13 further validate this (Note that the algorithm "Gemp et al., 2024 with the sampling method of Gemp et al." in Figure 13 is identical to the original algorithm design presented in (Gemp et al., 2024)). In fact, as detailed in Appendix D, all the loss functions used in our all experiments incorporate the stop-gradient operator.

---

> ### Author Response · Authors · 2024-11-23
>
> **Q2**: The input, parametrization and usage of NNs should be expanded further.
>
> **A**: The network architecture is described in detail in Section 5 Experiments: “The network, parameterized by ${\theta}$ and responsible for representing strategy profiles, is structured as a three-layer MLP. Both the input and hidden layers consist of 1024 neurons, while the output layer has $|\mathcal{N}|$ heads, where each head’s dimension corresponds to the action space of its respective player. The hidden layers utilize the ReLU activation function (Krizhevsky et al., 2012), and the output layer applies the Softmax activation function (Dempster et al., 1977), ensuring that the output resides within the simplex."
>
> Therefore, in the experiment, the neural network's input is a constant 1024-dimensional vector rather than a representation of the game itself. However, any input is feasible, and this setup is used in our paper primarily for demonstration purposes. For example, in solving real-world games, the input could be a natural language description of the game. Specifically, the natural language description can be processed by an Large Language Model (LLM), and the output from the embedding layer of the LLM can then be used as the input to the network that solves for the NE.
>
> The reason for using a deep neural network is its expressive power, allowing it to capture highly complex patterns and relationships that are difficult to model with traditional methods. Neural networks, especially deep architectures, can approximate arbitrary non-linear functions, making them well-suited for representing the strategic decision-making process in game theory. The ability of deep learning models to learn from raw data—without the need for handcrafted features—enables them to uncover intricate equilibrium strategies that might otherwise be overlooked. We use such a neural network demonstrate the power of DL for approximating the NE (we provide a detailed discussion on learning NE via DL in Appendix A).
>
> **Q3**: The paper lacks sufficient comparison/discussion to related prior work. E.g. [2-4]
>
> **A**: The loss functions in the works [2] and [4] you referenced both include the max operator. According to the analysis in (Gemp et al., 2024), the inclusion of the max operator generally leads to biased estimates under sampled play. Regarding work [3], based on Eq. (9) of [3], the final output strategy involves an exponential operation. Even if we obtain an unbiased estimate of $A_p$ (as defined in [3]), we cannot obtain an unbiased estimate of the strategy itself. In other words, work [3] also encounters biased estimation issues under sampled play. In conclusion, these works you mentioned all suffer from the biased estimation issues under sampled play. In contrary, our work proposes an efficient way to learn the NE with an unbiased estimate.
>
> In addition, these works investigate network architectures tailored for learning NE
> through DL. However, these architectures are unsuitable for solving real-world games, as they assume
> that the payoff matrix can be fully loaded into memory as input to the network. In real-world games,
> the payoff matrix is often too large to fit into memory, necessitating solutions based on sampling a
> subset of the matrix, referred to as sampled play. In this paper, we consider "sampled play". We are unable to compare with these works because they focus on the scenario where the payoff matrix can be loaded into memory, whereas our work considers the sampled play.
>
>
> **Q4**: Is Equation 3 the gradient of some precise loss function? If not, why is it appropriate to consider your approach as finding stationary points?
>
> **A**: Please see our responses to **Q1**.
>
>
>
> Additional References
>
> [1] Yongacoglu, Bora, et al. "Paths to Equilibrium in Games." *The Thirty-eighth Annual Conference on Neural Information Processing Systems*.
>
> [2] Grill, Jean-Bastien, et al. "Bootstrap your own latent-a new approach to self-supervised learning." *Advances in neural information processing systems* 33 (2020): 21271-21284.
>
> [3] Chen, Xinlei, and Kaiming He. "Exploring simple siamese representation learning." *Proceedings of the IEEE/CVF conference on computer vision and pattern recognition*. 2021.

---

> > ### Comment · Reviewer_BQo9 · 2024-11-28
> > **Response**
> >
> > I thank the authors for their response, however, I am still not satisfied with description of the NN. Quoting from the paper/above,
> >
> > "The network, parameterized by and responsible for representing strategy profiles, is structured as a three-layer MLP. Both the input and hidden layers consist of 1024 neurons, while the output layer has N heads, where each head’s dimension corresponds to the action space of its respective player. The hidden layers utilize the ReLU activation function (Krizhevsky et al., 2012), and the output layer applies the Softmax activation function (Dempster et al., 1977), ensuring that the output resides within the simplex."
> >
> > It seems that the NN outputs mixed strategy profiles. I.e. for each agent a mixed strategy, however, this is a trivial task where N*M numbers suffice where N is the number of agents and M the number of actions. Ever for large games this is a very small number that is not the size of the full payoff tensor (M^N). Why do we need a NN to encode strategy profiles?
> >
> > Quoting from your response you state
> >
> > "Therefore, in the experiment, the neural network's input is a constant 1024-dimensional vector rather than a representation of the game itself. "
> >
> > How is this vector generated? Is this the same fixed vector for across games you consider? The pseudocode in the paper talks about a game simulator. How it is this game simulator related exactly to your NN architecture?
> >
> > Overall, although I very much appreciate the approach I think that the coupling between the theoretical and experimental parts is still not fully fleshed out and thus I maintain my score.

---

> > > ### Author Response · Authors · 2024-11-30
> > >
> > > We thank the reviewer's comments. Below are our responses to your questions.
> > >
> > > **Q1**: It seems that the NN outputs mixed strategy profiles. I.e. for each agent a mixed strategy, however, this is a trivial task where N*M numbers suffice where N is the number of agents and M the number of actions. Ever for large games this is a very small number that is not the size of the full payoff tensor (M^N). Why do we need a NN to encode strategy profiles?
> > >
> > > **A**: First, the difficulty of learning the NE is related to the size of the payoff matrix ($n m^n$), rather than the size of $mn$, as the NE is defined by the payoff matrix. The following shows the game used in our experiments, where it is evident that Majority Voting has the smallest value of $mn$ but the largest payoff matrix size. As mentioned in our previous response to you, "the reason for using a deep neural network is its expressive power," it is intuitively clear that a larger game matrix requires a larger neural network to learn the NE defined by the payoff matrix.
> > >
> > > | Game                 | Payoff Matrix Size | Action Size | Players |
> > > | -------------------- | -----------------: | ----------: | ------: |
> > > | Kuhn Poker           |              8,192 |          64 |       2 |
> > > | Goofspiel            |            294,912 |         384 |       2 |
> > > | Blotto               |         75,898,944 |          66 |       4 |
> > > | Liar's Dice          |         10,616,832 |       2,304 |       2 |
> > > | Bertrand Oligopoly   |         25,000,000 |          50 |       4 |
> > > | Guess Two-Thirds Avg |         25,000,000 |          50 |       4 |
> > > | Minimum Effort       |        121,500,000 |          30 |       5 |
> > > | Majority Voting      |        537,109,375 |           5 |      11 |
> > >
> > >
> > >
> > > **Q2**: "Therefore, in the experiment, the neural network's input is a constant 1024-dimensional vector rather than a representation of the game itself. " How is this vector generated? Is this the same fixed vector for across games you consider?
> > >
> > > **A**: For all games, we use a 1024-dimensional vector, where all coordinates are set to 1, as the input. Furthermore, as previously mentioned, "any input is feasible, and this setup is used in our paper primarily for demonstration purposes. For example, in solving real-world games, the input could be a natural language description of the game. Specifically, the natural language description can be processed by an Large Language Model (LLM), and the output from the embedding layer of the LLM can then be used as the input to the network that solves for the NE."
> > >
> > >
> > >
> > > **Q3**: The pseudocode in the paper talks about a game simulator. How it is this game simulator related exactly to your NN architecture?
> > >
> > > **A**: The game simulator is not related to the NN. For all algorithms, whether tabular or NN based, the game simulator is essential, as it is required to return the reward of each player that can be obtained from the joint action ${a} \in \times_{i \in \mathcal{N}} \mathcal{A}_i$ (first introduced at line 130 in our paper) taken by all players.
> > >
> > >
> > >
> > > **Q4**: Overall, although I very much appreciate the approach I think that the coupling between the theoretical and experimental parts is still not fully fleshed out and thus I maintain my score.
> > >
> > > **A**: The primary motivation of our paper is to propose a lower variance unbiased loss function. In Section 4.2, we analyze the variance of our loss function, NAL, in comparison to the only known existing unbiased loss function (Gemp et al., 2024). Figures 2, 7, and 8 further validate our analysis, showing that the variance of NAL is significantly lower than that of other loss functions, whether biased or unbiased. Additionally, in Theorems 4.2 and 4.3, we demonstrate that by finding the stationary point of NAL, we can learn the NE. Figures 1, 4, 5, 11, 13, 14, 15, 16, 19, and 20 confirm our analysis and illustrate that the convergence rate of learning NE by finding the stationary point of NAL is much faster than that of existing loss functions.

---

### Author Response · Authors · 2024-11-23
**Main Modifications in Our Revision**

We sincerely thank all reviewers for their valuable feedback. We believe we have addressed your concerns. The main modifications in our paper are as follows (highlighted in red):

- We incorporate the stop-gradient operator to modify (i) the expression of NAL (Eq. (2)), and (ii) the implementation details of the loss functions used in the experiments (Appendix D).
- We remove the claim that “learning NE via NAL is simpler than minimizing other loss functions because NE is a stationary point of NAL while it is the global minimum of other loss functions.” We also revise our statement about tractability. For instance, we now state: “Inspired by the fact that learning a stationary point is simpler than finding a global minimum since a global minimum is necessarily a stationary point, while a stationary point is not always a global minimum, we ensure that an NE is a stationary point of NAL to improve computational efficiency.” In other words, in the revised version, we do not claim that learning NE via minimizing NAL is simpler than other loss functions.
- We add a new discussion on "Learning NE via DL" in Appendix A. This includes (i) why DL is used for leanring NE, and (ii) the limitations of existing DL-based NE learning algorithms.
- We include new experimental results, such as (i) results showing the performance of NashApr under a larger learning rate when the optimizer is SGD, and (ii) value curves of NAL during training.

---

### Author Response · Authors · 2024-12-02
**Reminder: Rebuttal Deadline Approaching for ICLR 2025 (Paper ID: 6366)**

Dear reviewers,

We are the authors of the paper (ID: 6366), "Learning Nash Equilibria in Normal-Form Games via Approximating Stationary Points." With the rebuttal period ending in less than a week, we would like to kindly ask if there are any further questions or clarifications you would like us to address. We are happy to provide any additional information to resolve any remaining concerns.

Best regards,

The Authors

---

### Meta-Review · Area_Chair_XPAx · 2024-12-18

**Metareview:**

The paper proposes a novel loss function, named Nash Advantage Loss (NAL). NAL is unbiased and exhibits significantly lower variance than the existing unbiased loss function (defined in a last year ICLR paper and is actually exploitability or Nash gap). It is also claimed that a Nash Equilibrium is a stationary point of NAL rather than having to be a global minimum, which improves the computational efficiency.
The AC and the reviewers believe that the latter statement is wrong or misleading: More specifically,

If Nash equilibria were stationary points of a smooth function, that would be a breakthrough and will imply collapsing of complexity classes (would imply CLS=PPAD). So NAL is a non-smooth function (or the claim is wrong). But if the function is non-continuous/smooth, how the authors claim that they improve computational efficiency from prior works? Getting a Nash equilibrium is computationally hard task (see Daskalakis et al). Moreover, where is the rigorous definition of stop gradient? The paper heavily is based on this and should appear somewhere. In conclusion, the paper has non-rigorous statements and we recommend rejection. If the paper was purely experimental without non-rigorous mathematical claims, the outcome might have been different.

**Additional Comments On Reviewer Discussion:**

The reviewers shared the same opinion as is written by the AC above, there are quite a few non-rigorous claims and the paper cannot be accepted on its current form.

---

### Decision · Program_Chairs · 2025-01-22

Reject